# Controlled irrigation suppresses methane emissions by reshaping the rhizosphere microbiomes in rice

Kenny J. X. Lau,[1] Ali Ma,[1] Bin Chen,[1] Maria Shibu Thankaraj Salammal,[1] Srinivasan Ramachandran,[1] Naweed I. Naqvi[1,2]

**ABSTRACT** The rhizosphere microbiomes of rice plants under conventional flood irrigation consist of highly complex consortia of microorganisms and, in particular, methanogens purportedly associated with methane emissions therein. Controlled irrigation has been proposed as a cultivation method of choice over continuous flooding to reduce water and fertilizer usage in an aerobic environment. However, a systematic understanding of the assembly and function of microbiota in the rhizosphere under drip and flood irrigation remains unclear. Using empirical analyses, we report a significant reduction in methane emissions in controlled irrigation compared to the flooded environment. Genotypic or varietal differences did not influence such methane emissions under conventional flooded cultivation of rice. Using metagenomic sequencing and computational analyses, we provide a deeper understanding of how drip irrigation or continuous flooding affects the root-associated microbiomes in rice. Rhizosphere soil from two different rice varieties, Huanghuazhan and Temasek rice, grown under drip or flood conditions in a greenhouse, was collected over 2 months post-transplantation for metagenomic analysis. Our results reveal that drip irrigation favors microbes involved in the nitrifying-denitrifying processes, while continuous flooding enriches for methanotrophs and methanogenic archaea. Syntrophic microbiomes associated with methanogenesis were significantly reduced in drip irrigation. Several keystone taxa were evident in the co-occurrence network model related to methanogenic, methanotrophic, nitrifying, sulfur-oxidizing and sulfur-reducing activities. Lastly, oxygen availability and redox potential were identified as key drivers that reshape rhizosphere microbiota and the associated metabolic functional differences observed between the two irrigation regimes, leading up to the microbial mitigation of climate impact.

**IMPORTANCE** Unlike previous studies in alternate wet-dry irrigation systems, this study characterized the rice microbiomes in a controlled drip irrigation setting where water levels were maintained at low levels and soil remained unflooded throughout the entire season in a greenhouse. A reduction of more than 90% in methane emissions was observed with drip irrigation compared to flood irrigation. A significant correlation was found between levels of methane emitted and *mcrA* gene copies detected, with a Pearson correlation coefficient $R$ of 0.77 and $P$-value of 2.3e − 10. Methanogens are highly abundant in continuously flooded rice soil and are significantly reduced in drip-irrigated soil. Metagenomic profiling indicates that the shifts in microbial diversity under drip irrigation favor nitrifying microorganisms and are likely influenced by increased oxygen availability due to higher soil redox potential.

**KEYWORDS** methane, greenhouse gas emission, rice, climate change, microbiomes, redox potential

**Peer Reviewer** Mirna Vázquez-Rosas-Landa, University of Texas at Austin, Austin, Texas, USA

Address correspondence to Naweed I. Naqvi, naweed@tll.org.sg, or Kenny J. X. Lau , kennylau@tll.org.sg.

The authors declare no conflict of interest.

See the funding table on p. 18.

Climate change is a significant threat due to the rising global temperatures and adverse weather events, such as typhoons, droughts, heatwaves, and floods around the world, affecting livelihoods of farmers and impacting crop yields (1). Methane is the most potent greenhouse gas arising from agriculture that is responsible for climate change. In traditional cultivation, rice is grown in flooded paddies to prevent weed growth, and such excessive irrigation methods lead to emission of high levels of methane as the standing water provides an anaerobic environment that favors the growth of methanogens, which feed on the decaying organic matter in soil (2). Therefore, to combat climate change, reducing methane emissions from rice fields has become one of the important endeavors in agricultural settings to achieve net zero by 2050.

Soil is rich in microorganisms due to the availability of diverse nutrient and carbon sources, and the diversity of microbes is thought to be important for maintaining soil health (3). The rhizosphere refers to the soil region around the plant roots and is a microenvironment where microorganisms thrive and interact with the host plants in a variety of ways that may be neutral, beneficial, or detrimental (4). Therefore, studying plant-soil microbiome interactions can provide further insights into how microorganisms can affect the environment and the health and productivity of a crop (5). Soil microorganisms drive nutrient cycling, breaking down organic matter and making essential nutrients like nitrogen and phosphorus available to plants. Understanding the soil microbiome can help mitigate climate change, as microbes regulate carbon sequestration and greenhouse gas emissions.

Rice (*Oryza sativa* L.) is an important staple food crop in Asia, Africa, and Latin America (6). The production of methane in soil is associated with anaerobic conditions caused by flooding. When deprived of oxygen, microorganisms utilize alternative inorganic electron acceptors like nitrate, ferric iron, and sulfate to decompose organic matter into substrates like acetate for methanogens (7). The crucial factor controlling which acceptor is used depends on the soil redox potential. Microorganisms favor the acceptor offering the highest energy yield, leading to a characteristic sequence. Reduction proceeds from oxygen with highest redox potential through nitrate, sulfate, down to the negative potential required for methanogenesis. Hence, studies have shown that disrupting this anaerobic, low-redox environment by draining the flooded soil might effectively reduce methane production (7, 8).

Different irrigation regimes, such as continuous flooding and drip irrigation, may have diverse effects on the rhizosphere microbiota in rice paddies. Traditionally, continuous flooding has been used for cultivation, wherein rice plants are grown in water at a depth of at least 10 cm to control weed growth (9). Therefore, recent studies have proposed alternative means such as controlled irrigation to save water and to mitigate methane production and emission levels (10). Drip irrigation methodology reduces water usage and aerates the soil to minimize anoxic conditions and is deemed effective in reducing methane emissions.

In this study, we assessed and characterized the soil microbiomes of two indica rice varieties, Huanghuazhan (HHZ, China) and Temasek rice (Singapore), under drip irrigation and continuous flooding conditions in a greenhouse. The microbiome composition and structure helped provide further insight on the possible microbial activities and metabolic functions in the soil. In addition, potential interactions among different microbial species/taxa were inferred using microbial co-occurrence network analyses (11, 12). Important keystone members were identified to predict microbial clusters and their relation to methane emissions. HHZ is an elite indica rice variety from South China that is known for its high yield (13). Temasek rice is a variety bred using marker-assisted breeding techniques performed in Singapore (14). Temasek rice also has high yield, disease resistance, and submergence-tolerance traits that impart adaptation to various cultivation regimes and environmental conditions. Using shotgun metagenomics analyses, we (i) unravel the taxonomic and functional shifts in the root microbiomes under flood- and drip-irrigated conditions, (ii) identify syntrophic microbial communities that are involved in methane production and tropism, and (iii) provide

insights into nutrient metabolism and its influence on methane emissions and/or suppression. Lastly, we compared the root morphology under flood and drip-irrigated rice systems together with a detailed analysis of the soil parameters such as oxidation reduction potential (ORP) and pH associated with anaerobic and aerobic cultivation regimes.

## MATERIALS AND METHODS

### Methane emission measurements

Methane ($CH_4$) emissions were captured across the important growth stages of the rice plants. The closed chamber method was used to measure methane, where transparent chambers (50 cm × 50 cm × 50 cm) equipped with an electric fan and a sampling window were placed over the plastic trays such that four rice plants were included in one chamber (15). A chamber base was installed in the soil, and sampling was performed from the beginning of the season to the end between 0930 and 1230 h every week, and gas samples were collected from the top chamber using a 20 mL syringe at 0, 10, 20, and 30 min during the second week post-transplantation. $CH_4$ measurements were then analyzed using a gas chromatograph equipped with a flame ionization detector. Methane flux was determined using the following formula:

$$F = \rho \times \left( \frac{V}{A} \times \frac{\Delta c}{\Delta t} \times \frac{273}{K} \right)$$

$F$ is $CH_4$ flux expressed in mg $CH_4$ m$^{-2}$ h$^{-1}$, $\rho$ is the $CH_4$ gas density (0.174 mg cm$^{-3}$), $V$ is the volume of the chamber in (m$^{-3}$), $A$ is the surface area of the chamber (m$^{-2}$), $\frac{\Delta c}{\Delta t}$ is the rate of gas concentration increase in the chamber (mg m$^{-3}$ day$^{-1}$), and $K$ is the air temperature in the chamber measured in Kelvin.

GC-FID-based analysis was performed using the column Poraplot Q-HT, 25 m, with carrier gas helium at a flow rate of 18 mL/min set at column temperature of 90°C, FID at 150°C, hydrogen at 40 mL/min, and air at 400 mL/min. Results were plotted as box plots using the ggplot2 (16) package in R.

### Experimental design and soil sampling

The soil mixture was made by amending topsoil with 30% (vol/vol) peat moss (BVB, Netherlands), sheep manure of 300 g/m$^2$, and finely chopped rice straw of 1,000 g/m$^2$ and was then thoroughly blended and placed in plastic tubs of 0.99 m (L) × 0.67 m (W) to a depth of 30 cm. The trays were filled with water until the soil was fully submerged and left for two weeks to facilitate the decomposition of the added straw. After the incubation period, 15 g of triple superphosphate was added to each tray. Twenty-day-old seedlings of HHZ and Temasek rice were transplanted with a spacing of 25 cm between each plant in the greenhouse facility at Temasek Life Sciences Laboratory, Singapore (103° 46′ 36″ E and 1° 17′ 36″ N). The two rice varieties were then tested with three seedlings. Each tray was then subjected to different irrigation treatments, drip and flood. Drip-irrigated trays had drainage holes at the bottom to allow excess water to escape. The drip-irrigated trays were laid with two lateral drip irrigation lines between the rows to provide water before planting. Watering was carried out through the drip system, including the application of fertilizer, for 2 min per cycle, repeated four times daily after planting. In contrast, in the continuously flooded trays, the rice plants were submerged in water to a depth of 1 inch above the soil level before planting. Fertilizer was applied according to standard agronomic practices. The soil samples were processed for metagenomic analysis during the last week of harvest.

Soil samples were collected using 25 mL serological pipette strips that were modified as soil probes (Fig. S1a). Soil sampling was performed in duplicates for each variety and irrigation regime. Using a heat-sterilized knife, the serological pipettes were cut between the 16 and 20 mL marks to create an opening at the top and the side (Fig. S1b and c). A

new 10 mL serological pipette was then inserted from the top of the 25 mL pipette strips to enclose the 25 mL pipette strip (Fig. S1d and e). The soil probe construct was then plunged into the soil adjacent to the stem of the rice plant at a depth of 15 to 20 cm, and the 10 mL strip was then released so that soil can enter the 25 mL strip chamber from the side opening (Fig. S1f). Soil samples were then collected in a 50 mL tube, transported to the laboratory, and kept frozen at −20°C freezer until DNA extraction.

## DNA extraction and quantitative PCR

One gram of soil was used for DNA extraction using the DNeasy PowerSoil kit (Qiagen, Germany) following the manufacturer's standard protocols. DNA was eluted in 100 µL standard Tris-EDTA buffer and stored at −20°C freezer until processed for nucleotide sequencing. Quantitative polymerase chain reaction (qPCR) was performed using primers targeting the *mcrA* gene. Primer pair used was Met630F: GGATTAGATACCC SGGTAGT and Met803R: GTTGARTCCAATTAAACCGCA. The qPCR mixture consisted of 1 µL of DNA template and 0.25 µM of primers with 2× master mix. Negative controls were tested using sterile water. qPCR was performed using the KAPA SYBR Fast Universal qPCR kit (Roche, USA) in a CFX96 thermal cycler (Bio-Rad, USA). The standard curve for qPCR analysis was prepared by serial dilution of a gBlock containing known $10^{10}$ copies of the *mcrA* gene of *Methanobacterium,* as follows: CCA GGG GCG CGA ACC GGA TTA GAT ACC CGG GTA GTC CTG GCC GTA AAC GAT GCA GAC TTG GTG TTG GGA TGG CTT CGA GCT GCT CCA GTG CCG AAG GGA AGC TGT TAA GTC TGC CGC CTG GGA AGT ACG GTC GCA AGA CTG AAA CTT AAA GGA ATT GGC GGG GGA GCA CCA CAA CGC GTG GAG CCT GCG GTT TAA TTG GAT TCA ACG CCG GAC synthesized by Integrated DNA Technologies (IDT, USA). Gene copy number abundance was determined based on the straight-line equation of Cq versus log of known copies of the standard curve template. The gene copy numbers were then normalized to 1 g of soil.

## Library preparation, sequencing, and bioinformatics analysis

Extracted DNA samples were quantitated using a Nanodrop spectrophotometer (Thermo Fisher Scientific, USA). High-throughput sequencing libraries were prepared with the VAHTS Universal Plus DNA library preparation kit for Illumina V2 ND627 (Vazyme, China) following manufacturer's protocol. The samples were sequenced at 1st BASE/ Axil Scientific, Singapore, using the NovaSeq 6000 sequencer (Illumina, USA). Shotgun metagenomic sequencing was performed. Sequence adapters were first trimmed from all paired-end reads using bbduk of the BBTools packages (17). MultiQC (18) was performed and reads with at least a quality Phred score of 20 were filtered. The filtered reads were then assembled *de novo* using Megahit (19) into contigs of at least 500 bp. Ribosomal RNAs were predicted using Barrnap (20) and classified using RDP classifier (21). Open reading frames on the assembled contigs were predicted using Prodigal (22) and then queried across the updated NCBI nr database using Diamond v.2.0.8.146 (23) and HMMER3 (24). The output was then visualized in Krona (25) and Metagenome Analyzer v.6.24.22 (26).

The taxonomic tables at genus and class levels were exported for subsequent statistical analysis in R using vegan (27) and phyloseq (28) packages. Alpha and beta-diversity were both assessed using the Bray-Curtis dissimilarity metric. Alpha-diversity was analyzed using Chao-1, Shannon (H), and Simpson (D) diversity indices. Taxa were collapsed at the class level and imported into R. The microbiome of top 10 classes was plotted as stacked bar charts using the plotly package (29). Canonical correspondence analysis (CCA) plots were created to observe clustering patterns in relation to drip and flood irrigation. Differential taxa analysis was performed using the MaAsLin2 package (30) with drip and flood as fixed factors using default parameters in R. The co-occurrence network analysis was performed using a R script adapted from Li et al. (31). Methane and ORP measurements were appended to the correlation matrix. The co-occurrence network was visualized using Gephi v.0.9.2 (32). To simplify the network graph, the top 200 genera were filtered and used. Using a cut-off score of Spearman's rho 0.6, *P*-value

of 0.01, and clustering algorithm in Gephi, a network model consisting of 134 nodes and 391 edges was obtained. Reads collapsed at the genus level with taxonomic rank hierarchy information were exported from MEGAN and converted to biom format. The biom tables were then queried for potential metabolic functions in Functional Annotation of Prokaryotic Taxa (FAPROTAX) (33). The output file was converted to a tab-delimited file and plotted as a stacked bar chart in relative abundances. Putative metabolic functions of microbes that are present in drip and flood-irrigated soil were predicted by mapping the metagenomes to putative functions in the Kyoto Encyclopedia of Genes and Genomes (KEGG) database. Reads were expressed as RPKM (reads per kilobase per million mapped reads) and normalized as Z-score in the heatmap. The raw counts of KEGG annotations were also exported together with the sample metadata for analysis in MaAsLin2 for differential gene function analysis.

## Soil redox potential measurements

A portable pH, ORP, and temperature meter HI991003 (Hanna Instruments, Italy) was used to measure the soil redox potential. A hole of 15 cm–20 cm depth was made using modified soil probes before inserting the ORP meter probe. The ORP probe was then left in the soil for 5 min to allow readings to become stabilized before recording them.

## Root size measurements

HHZ and Temasek rice plants were uprooted first by loosening the soil with running tap water, followed by a spade at the end of the growing season. Triplicates of rice plants of each species were chosen at random from drip and flood irrigation trays. Plants were laid side by side on a black canvas, with drip-irrigated plants on the left and continuously flooded plants on the right, and a 30 cm ruler in the middle. The images of the plant roots were captured and imported into ImageJ (34). A 30 cm ruler was used as a reference, and measurements were converted from pixels to centimeters to derive the actual root length of each plant.

## RESULTS

### Methane emissions are significantly reduced in drip-irrigated rice cultivation

Methane emissions were measured using the static closed chamber method, with gas emissions measured in the range of 0 to 500 mg/m$^2$ per day. Based on our findings, methane emissions from rice soils under continuous flooding generally showed two emission peaks throughout the life stages of the rice plants (Fig. 1a). The first peak occurred at week 4 post-transplanting during the seedling to tillering stage, while the second peak coincided with the transition from tillering to flowering stage at week 8. A separate study also showed a similar two-peak methane emission profile in flooded rice soil. Interestingly, they also found that for the same rice variety when planted at three different locations, there may be two different methane emission patterns depending on the levels of organic matter added in the soil at the different stages of rice development (35). In contrast, drip-irrigated rice soils seemed to emit negligible levels of methane as compared to their continuously flooded counterparts, and we observed a statistically significant 96%–98% reduction ($P < 0.05$) in methane emissions in such controlled drip irrigation regime for the entire season (Table S1). No significant differences were observed between HHZ and Temasek rice varieties, suggesting a lack of the genotype-based contribution to methane emissions.

To track the source of methane emissions, the methyl coenzyme M reductase (*mcrA*) alpha subunit can be used, and it is a well-established gene marker thought to be highly conserved and highly specific for methanogens. Several studies have used this gene as a biomarker to estimate the methanogenic profile in rice paddies, peat soil, and anaerobic digesters (36–38). Similarly, quantitative PCR (qPCR) was also performed in our experiment to correlate *mcrA* gene copy numbers with methane flux levels. Our results yield a strong correlation of $R = 0.77$, $P = 2.3e - 10$ was observed between methane flux and

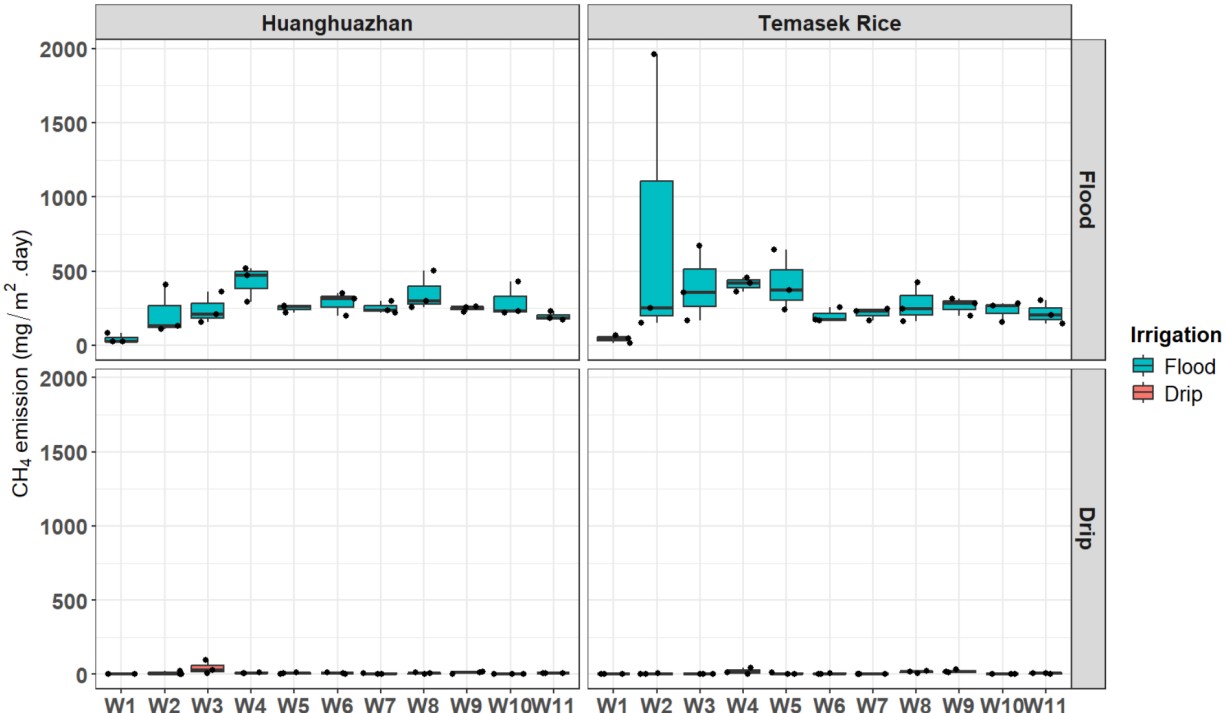

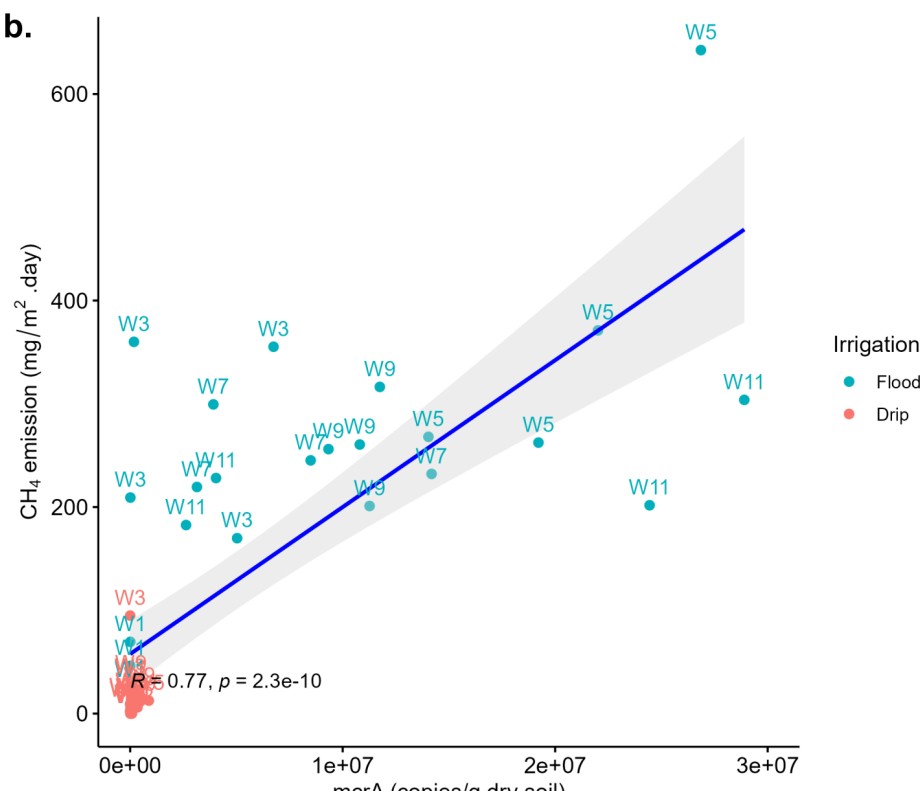

**FIG 1** Drip irrigation reduces methane (CH$_4$) emissions in rice compared to continuous flooding regime. (a) Boxplot showing methane emission levels across 11 weeks of rice cultivation, where the top panel shows the continuous flooding regime and the bottom panel displays the drip irrigation regime. Flux rates are expressed in mg CH$_4$ m$^{-2}$ day$^{-1}$. (b) Methane readings were correlated with *mcrA* gene copy number estimates based on quantitative PCR readings. *mcrA*

Fig 1 (Continued)

gene copies of flood-irrigated samples were highly correlated with methane readings with a Pearson correlation coefficient *R* of 0.77 and *P*-value of 2.3e − 10. Drip-irrigated soil, on the other hand, did not show any correlation.

*mcrA* gene copy per gram dry soil in the continuous flooded soil samples (Fig. 1b). However, such correlation was not evident in drip-irrigated samples, as methane emissions were near or below detection limits and gene copy number counts were significantly lower (Table S2). These findings highlight that drip irrigation can be used as a methane mitigation strategy in rice cultivation, since it shows a significant reduction in greenhouse gas emissions, in particular methane, as compared to traditional continuous flooding. The consistent methane reduction across both rice varieties shows the potential of irrigation management practices to address the effect on methane emissions.

## Drip irrigation affects the root-associated microbiomes

Shotgun metagenomics sequencing of soil samples was performed to investigate how drip irrigation impacts the total microbial community structure in the soil and the corresponding methane emission levels. Duplicate samples were taken for metagenomics analysis from drip and flood-irrigated rice varieties HHZ and TR. An average of 51,176,651 mapped reads were obtained. A summary of sequence reads obtained from each sample is presented in Fig. S2 and Table S3. Contigs and taxonomic binning information can be found in the supplementary data and at https://datadryad.org/dataset/doi:10.5061/dryad.76hdr7t8s. The metagenomic profiles of the soil microbial samples were normalized and analyzed as relative abundances. The results showed that drip and flood-irrigated soils have different microbial community structures and diversity. In terms of differences in species diversity (alpha-diversity), the drip-irrigated soil was found to be less diverse than flooded soil (*P* < 0.01) and showed fewer numbers of microbial species than the ones observed in flooded soils (Table S4). There was no significant difference in Chao1, Shannon, and Simpson indices between rice varieties as both HHZ and Temasek rice (TR) showed similar scores and their associated microbiota abundance was not considered to be statistically different from each other (Table S5). Principal coordinate analysis (PCoA) of Bray-Curtis distance showed two distinct clusters of drip and flood-irrigated soils (Fig. S3).

Notable differences were evident at the class level, in the proportions of various taxa across drip and flood-irrigated soil. *Gammaproteobacteria*, *Cytophagia,* and *Nitrospira* were higher in abundance in drip-irrigated soil (Fig. 2). On the other hand, *Methanobacteria*, *Actinomycetia,* and *Methanomicrobia* were highly abundant in the flooded soil. *Actinomycetia* and *Proteobacteria* are potential decomposers of complex substrates and are essential for carbon cycling (39). *Actinomycetia* are comprised of r-strategist bacteria that grow fast and contribute to biologically active metabolites in the soil (40). For instance, *Dactylosporangium* and *Streptomyces* were found to enhance organic decomposition and promote nitrogen fixation and plant growth (41, 42). At the genus level, microbes can be classified into functional guilds such as nitrifiers, methanogens, methanotrophs, nitrogen-fixing, sulfur-oxidizing, sulfur-reducing, and syntrophic microbial communities (Fig. 3). In the drip-irrigated soil, an increase in nitrifiers with a corresponding decrease in methanogens and methanotrophs was observed as compared to the continuously flooded soil. Specifically, within the Archaea domain, methanogens such as *Methanothrix*, *Methanosarcina*, *Methanocella,* and *Methanobacterium* were higher in abundance in flooded soil, while *Candidatus Nitrosocosmicus* showed greater abundance in the drip soil (Fig. 4 and 5). CCA showed a significant separation between drip and flood samples along the first CCA axis with 58.6% of variance explained, where *Nitrospira*, *Bradyrhizobium,* and *Pseudomonas* are driving the change in drip-irrigated soil (Fig. 6). Under continuous flood irrigation, soil microbiome profiles associated with HHZ and TR varieties exhibited close clustering, indicating compositional

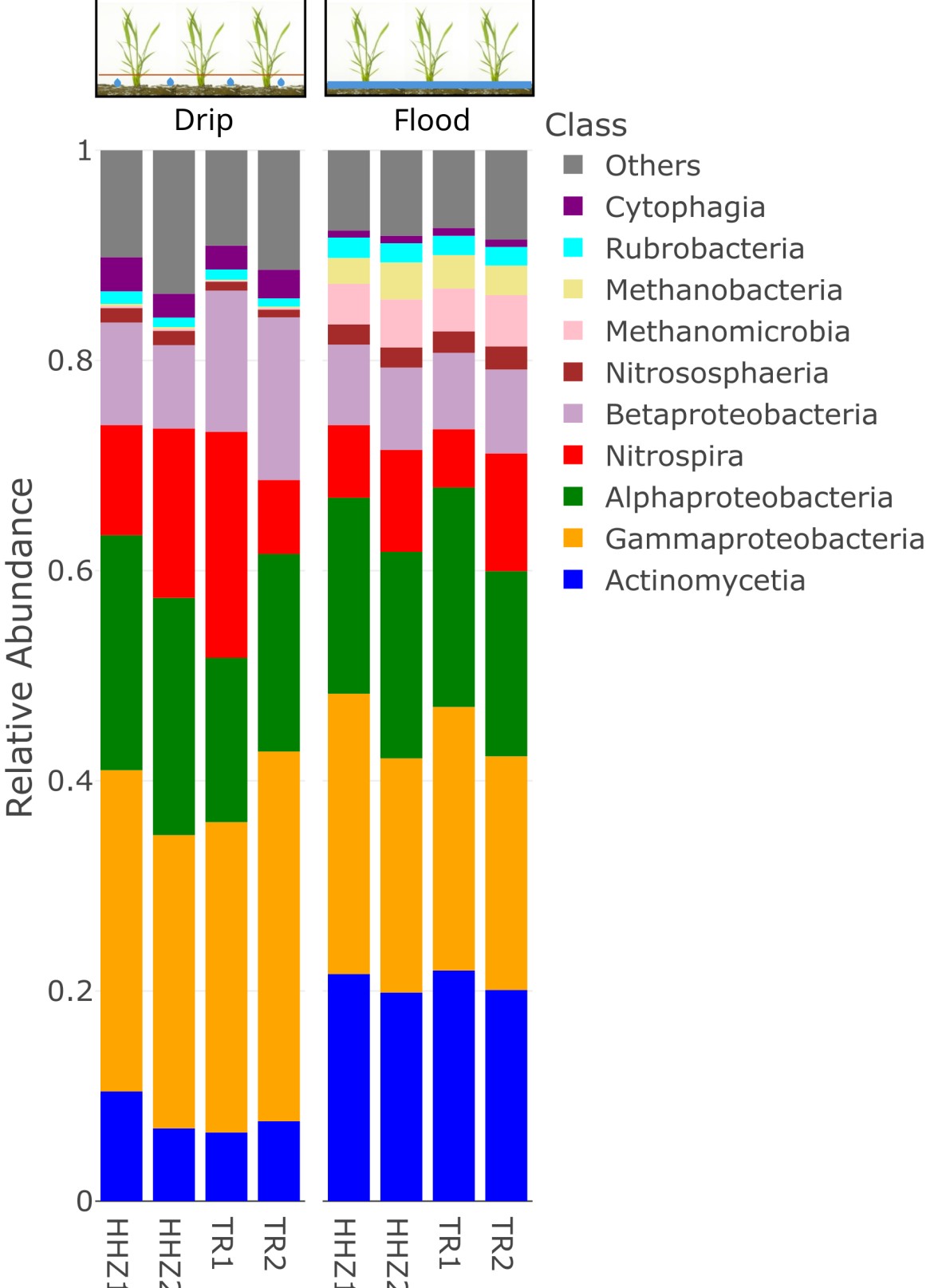

**FIG 2** Soil microbial community composition at the class level for drip-irrigated versus flooded rice. The stacked bar chart displays the community composition of the top 10 most abundant taxa collapsed at the class level. Each stacked bar with a different color represents a class-level taxon and its corresponding relative abundance level under drip and flood irrigation conditions.

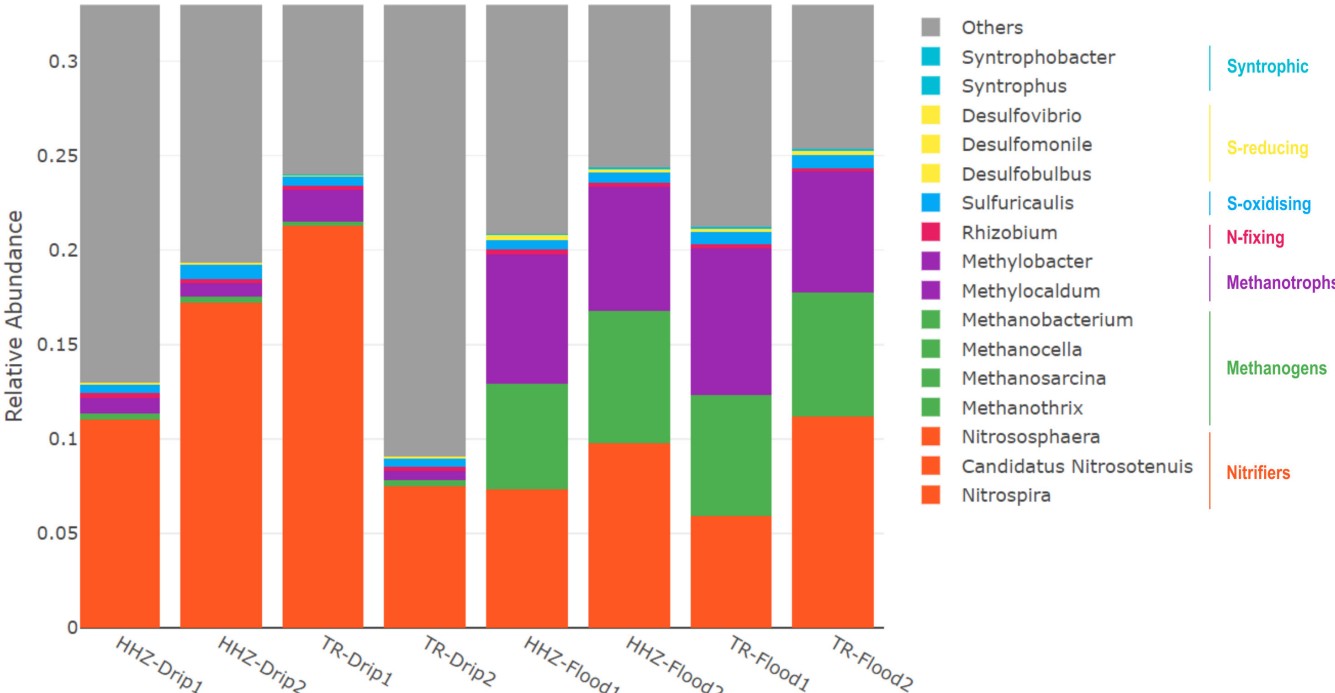

**FIG 3** Drip and flood irrigation reshapes key microbial taxa in soil. The stacked bar chart displays community composition of taxa colored by their functional group. Seven functional categories, nitrifiers, methanogens, methanotrophs, N-fixing, S-oxidizing, S-reducing, and syntrophic, were selected. *Nitrososphaera*, *Candidatus Nitrosotenuis,* and *Nitrospira* are highly abundant, and they contribute to about 1% of the microbiome in red. This is followed by methanogenic archaea in green comprising *Methanobacterium*, *Methanocella*, *Methanosarcina,* and *Methanothrix,* as well as Methanotrophs in purple consisting of *Methylobacter* and *Methylocaldum*. Duplicate samples were taken for each irrigation condition from the two rice varieties.

similarity. In contrast, samples from drip irrigation treatment displayed greater heterogeneity, with partial separation observed between the rice varieties or genotypes. *Methylocaldum*, *Nocardioides*, *Hyphomicrobium,* and *Lysobacter* were identified as key taxa that shape the continuous flooded soil microbiomes. The drip-irrigated soil was found to be enriched in nitrifiers (*Nitrospira*) and denitrifiers (*Pseudomonas*) when rice

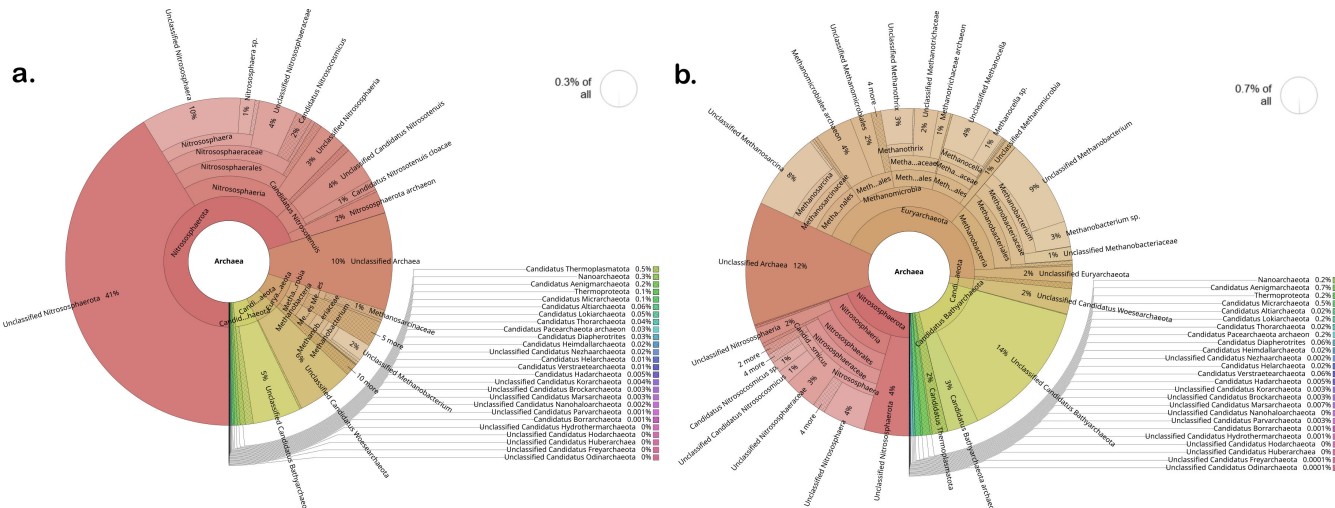

**FIG 4** Archaeal community structure in HHZ rice soil under two irrigation treatments. Krona charts show the relative abundance of archaeal taxa in (a) drip-irrigated and (b) flooded soils where brown and khaki color represents methanogens. The archaeal community under drip irrigation accounted for approximately 0.3% of the total soil microbiome. In contrast, the flooded condition selected for a methanogen-rich community.

plants are grown aerobically, while the continuous flooding environment selected for methanotrophs such as *Methylocaldum*, *Nocardiodes,* and *Actinomycetes* like *Streptomyces* and *Agromyces*. The drip-irrigated soil also tended to be more heterogeneous as compared to the flooded soil, where flooded soil samples were found to cluster closely together along CCA2 axis.

Further analysis was performed using MaAsLin2 to investigate which microbes are enriched in each environment. Our results reveal a list of taxa that are statistically significant in either flood or drip-irrigated environments (Supplementary Data). Taxa that were significantly enriched in flood-irrigated rice soil include *Methanothrix*, *Methanocella*, *Methanosarcina,* and *Methanobacterium* (Fig. S4). Drip-irrigated soil, on the other hand, selects for *Nitrospira*, *Candidatus Nitrosotenuis*, *Nitrococcus,* and *Nitrospirillum*. In flooded anaerobic environments, such as rice paddies, methanogens thrive due to the absence of oxygen and the presence of organic matter as a substrate. These specialized microorganisms play a role in the carbon cycle by breaking down complex organic compounds into methane, a process driven by anaerobic conditions. In contrast, de-nitrifiers and nitrifiers seem to be enriched in drip-irrigated aerobic environments where oxygen is readily available. Nitrifiers, such as ammonia-oxidizing bacteria, for instance *Nitrospira*, facilitate the conversion of ammonia to nitrites and subsequently to nitrates, while denitrifiers like *Pseudomonas* and *Hydrogenophaga* utilize nitrate as an electron acceptor under microaerophilic conditions, converting it into gaseous nitrogen. The stratified oxygen availability between these environments creates distinct niches, enabling the coexistence of these microbial groups with complementary roles in the global biogeochemical cycles.

Metabolic functions of microbes that are present in drip- and flood-irrigated soil were predicted, and the heatmap shows higher Z-scores for methane-related genes such as tetrahydromethanopterin-S-methyltransferase (*mtrA-mtrH*) and methanol dehydrogenase that are key enzymes in the methane biogenesis pathways found in methanogens like *Methanobacterium* and *Methanosarcina* (Fig. S5). In addition, KEGG gene function abundances were also analyzed in MaAsLin2. Since archaea only make up less than 1% of the total microbiome, many of the inferred functions may be mapped to activities of the bulk soil rhizosphere bacteria rather than methanogens. From the KEGG and MaAsLin2 analysis, the association plot revealed that there is greater upregulation of nutrient transporter and energy metabolism pathways under flood conditions, whereas drip irrigation was associated with increased abundance of genes involved in cellular repair and stress response (Fig. S6). No significant gene functional differences were detected between HHZ and TR rice varieties.

Additionally, putative metabolic functions were also predicted using FAPROTAX. Functional activities relating to methanogenesis, methanotrophy, methylotrophy, methanol oxidation, hydrocarbon and aromatic compound degradation were identified and were higher in continuously flooded soil (Fig. S7). These functions further show that anaerobic degradation and methanogenesis activities are increased in flooded soil and are, consequently, associated with the higher methane emissions and flux in flooded rice cultivation. It supports the hypothesis that anaerobic degradation of complex substrates from organic matter occurs upstream to produce acetate that feeds downstream processes like methanogenesis, which in turn produces methane that acts as a substrate for methanotrophs. Results also suggest that a complex microbial community network exists between these functional groups of microbes. In contrast, aerobic chemoheterotrophy was found to be higher in drip-irrigated soil, and this observation was in line with the higher redox state due to aerobic conditions under controlled irrigation. No significant differences in FAPROTAX functions were observed between the microbiomes associated with the HHZ and TR rice roots for both drip and flood conditions.

We also examined the root morphology of both rice varieties under different irrigation conditions. Our results showed that drip-irrigated rice plants tend to have shorter roots as compared to those in continuously flood-irrigated plants (Fig. S8; Table S6). In addition to shorter lengths, the roots of the drip-irrigated rice plants showed

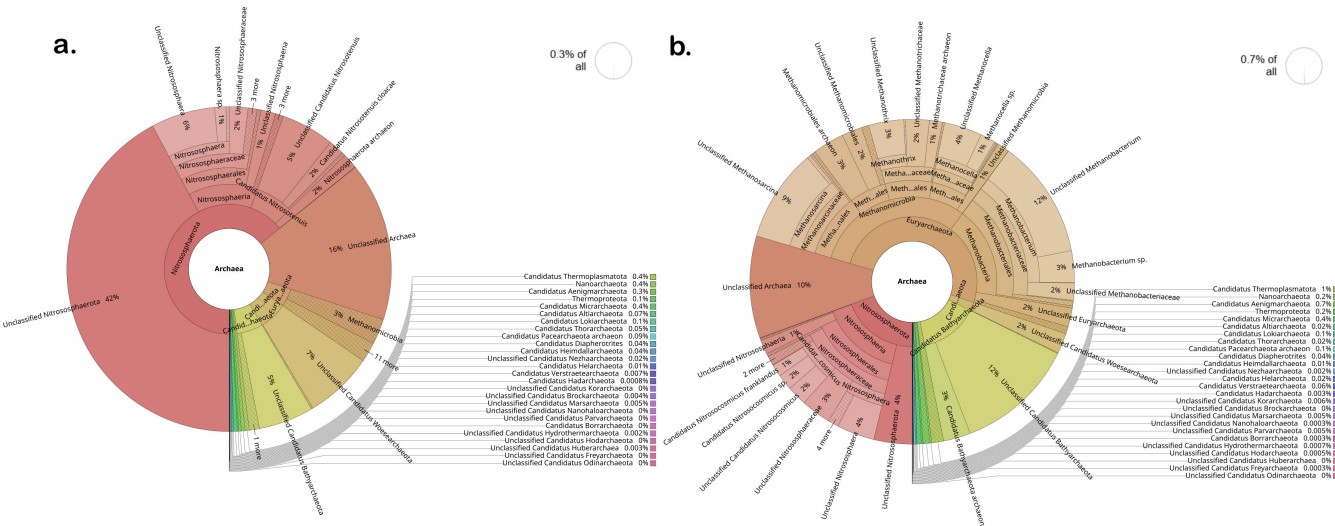

**FIG 5** Relative abundance of archaeal population in Temasek rice soil microbiomes under drip and flood conditions. Krona charts illustrate the relative abundance of archaeal taxa in (a) drip-irrigated soil and (b) flooded soil, where brown and khaki colors represent the methanogens. The archaeal community was reduced under drip irrigation, which accounts for 0.3% of the total microbiome. Under flooding, total archaeal abundance increased to 0.7%, with significant enrichment of methanogens.

reduced lateral growth and were thicker in diameter. The observations were consistent and similar in both HHZ and Temasek rice.

## Redox potential is higher in drip-irrigated than continuous flooded soil

ORP was also measured across the different growth stages of the rice plants over the entire season (Fig. 7). Drip-irrigated soils tend to be more heterogeneous and had higher variance in the ORP readings with a median value of 111.5 mV. Conversely, the flood-irrigated soils yielded readings that are in the negative range with a median of −205.5 mV. Drip-irrigated soil tends to exhibit a more positive redox potential, typically around 100 mV, due to its aerobic conditions facilitated by controlled water delivery and oxygen diffusion.

## Identification of co-abundance network of methane biogenesis and ORP-related clusters

Molecular ecological networks were created to unravel pairwise correlations between methane emissions and ORP measurements with microbe-microbe interactions among the top 200 taxa. After applying a Spearman rank correlation cutoff of 0.6, the final network graph is undirected and consisted of 134 nodes and 391 edges. The network has a diameter of 12, an average path length of 4.18, and an average degree of 5.836. The average path length represents the average shortest distance between all possible pairs of nodes, whereas the average degree represents the average number of connections each node has with other nodes in the network. Hence, this model is considered sparse as each microbe generally has an average of five to six neighbors. The network, however, boasts a modularity index of 0.671, suggesting that it has a moderate propensity for species nodes to form densely interconnected subgroups despite the sparse number of neighboring nodes. The co-occurrence analysis revealed five primary clusters where each cluster is subsequently labeled according to a common putative function that the members likely performed. For instance, *Methanocella*, *Methanoculleus,* and *Methanoregula* form part of the methane and methanol cycling cluster, and the relative abundances correlate strongly with methane flux, while *Nitrospira*, *Candidatus Nitrosotenuis*, *Nitrococcus,* and *Nitrospirillum* are part of the facultative anaerobe-redox potential cluster that is associated with higher redox potential in drip-irrigated soil.

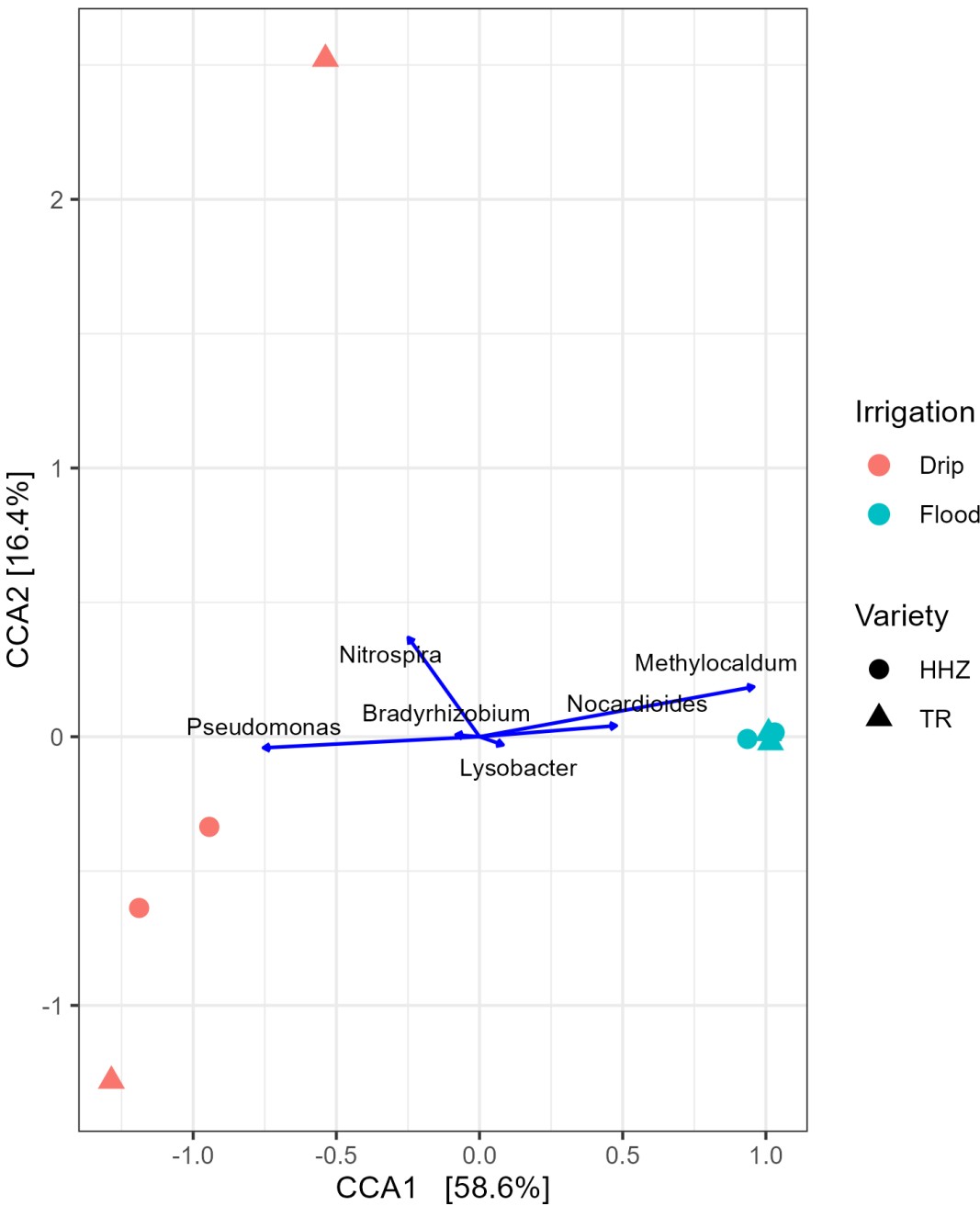

**FIG 6** The CCA plot shows differential genera groups in the rhizosphere microbiomes under drip and flood-irrigated conditions. Samples separate distinctly along the first canonical axis (CCA1), which explains 58.6% of the constrained variance. Samples from the continuously flooded treatment form a cluster, indicating low compositional variation, whereas samples from the drip-irrigated treatment exhibit greater dispersion. Key genera are represented as vectors and are overlaid to find differential taxa that drive the microbial changes in the flood- or the drip-irrigated samples. The second axis (CCA2) explains an additional 16.4% of the constrained variance accounting for the heterogeneity in the microbiomes between drip samples.

## DISCUSSION

### Methane emission profiles in rice soils are associated with the soil microbiome and water management practices

Methane flux modeling in rice fields typically shows four hypothetical temporal patterns based on data collected from 232 *in situ* rice field sites, aimed at guiding mitigation strategies in irrigation, organic matter, or fertilizer management (2). Among these

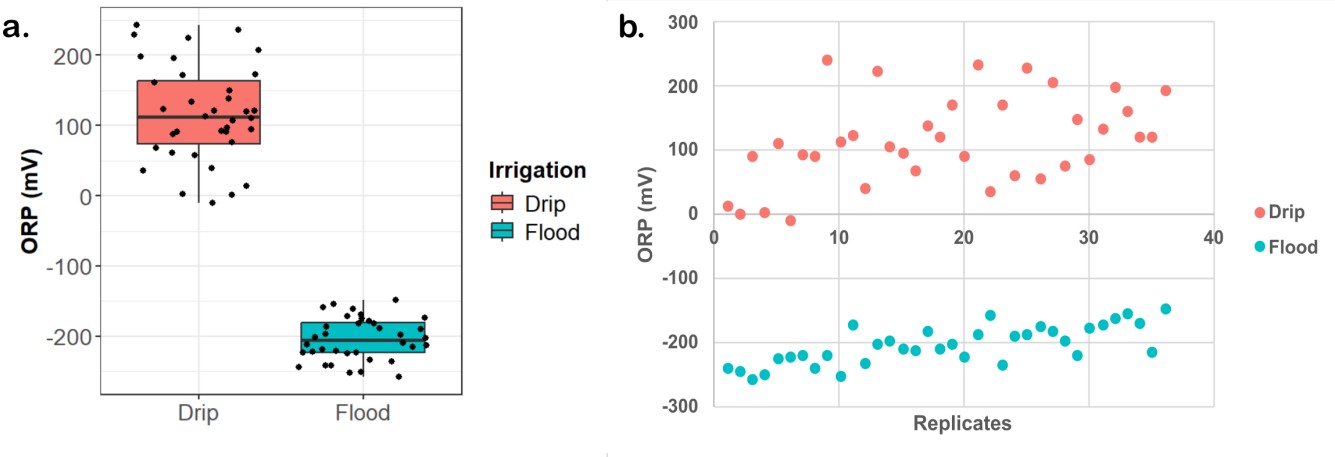

**FIG 7** Drip irrigation maintains a higher soil redox potential (ORP) compared to continuous flooding. (a) Box plot showing ORP measurements that were collected over an 11-week experimental period. (b) Scatter plot representations of the field readings reveal characteristic ORP (mV) under drip irrigation and continuous flooding. Out of the 72 measurements, 36 were flood and 36 were drip, and they were taken in duplicate.

models, the methane emission data for continuously flooded HHZ and Temasek rice, presented in Fig. 1a, aligns with the continuous emission model. As shown in Fig. 1a, methane emissions for both Temasek and HHZ rice peaked early during the tillering stage, specifically at week 4 post-transplantation. This observation also corresponds with the CH4MOD model by Hu et al. (43), which similarly illustrates a unimodal emission pattern under continuous flooding conditions (43). Additionally, a separate study also conducted a trial in the rice paddies of central Japan highlighted the potential of water management practices, such as alternate wetting and drying (AWD), as a strategy to effectively mitigate greenhouse gas emissions by up to 55%–60% (44). Similarly, our findings suggest that adopting strategies like drip irrigation can also make rice cultivation more sustainable. Overall, our results emphasize the critical role of irrigation techniques in shaping methane emission patterns in rice cultivation. The alignment of our empirical data with established emission models further validates the reliability of these findings, while the significant reductions demonstrated by alternative practices such as drip irrigation and AWD highlight their potential as viable climate-smart solutions. Our results showed strong correlation between *mcrA* gene copy number abundance and methane emissions ($R = 0.77$), and the dispersion of data points around the regression line suggested substantial variability across individual samples and data points. While *mcrA* gene abundance (qPCR) and metagenomic sequencing served as a useful proxy to detect methanogens, such DNA-based methods do not report the metabolic activity of the microbes *per se*. For instance, Qi et al. (45) used *mcrA* gene copy number and methane emissions data to evaluate the effectiveness of methane mitigation in both chemical fertilizer and biochar treatments (45).

We also observed changes in root morphology in drip-irrigated rice plants as shown in Fig. S7 and the phenotype could be attributed to the plant's response to water availability under different water management regimes. In drip irrigation, water is supplied in a controlled manner directly to the soil near the plant roots, reducing the need for extensive root exploration to access water. Our results suggest that the drip-irrigated plant roots tend to grow closer to the soil surface as water becomes scarce with increasing depth. The roots also tend to be thicker near the soil surface, which is likely an adaptation to absorb more moisture under such controlled irrigation regimes. On the other hand, the flood-irrigated rice plants seemed to develop longer and deeper roots with fine lateral growth. It is likely that the plant invests in a deeper root system to anchor effectively and to absorb more nutrients via the active microbial communities under excess water.

## Microbial communities are reshaped by changes in redox potential of the soil

Notable difference in soil redox potential was observed, with drip-irrigated soils exhibiting 111.5 mV and flooded soils showing −205.5 mV ORP. This change in redox potential likely correlates with a shift in the microbiome, influenced by the distinct electron donor and acceptor changes in the drip and flood environments. Soil redox potential between −100 and −200 mV has been associated with methane emissions in flooded rice soils (46). In fact, another study has shown that oxygen availability in the soil controls the soil redox potential (47). Our experiments showed that drip irrigation increases the redox potential significantly. Therefore, we observed a significant reduction in methane emission under controlled irrigation. Drip irrigation condition promotes oxidative processes, supporting the activity of aerobic microorganisms and processes like denitrification, where nitrate is converted into nitrogen gas. In contrast, continuously anaerobic flooded soil has an average redox potential of approximately −200 mV, indicative of strongly reducing conditions. These conditions arise from water saturation, which limits oxygen availability and promotes the activity of anaerobic microbes, such as methanogens, that rely on alternative electron acceptors like carbon dioxide. The difference in the redox potential reflects the contrasting oxygen availability and microbial dynamics within these two soil profiles. Therefore, understanding the soil redox dynamics will be important for optimizing water management practices as it shapes the microbial communities and influences greenhouse gas emissions.

The two groups of redox reaction pairs that are likely involved in this process are (i) aerobic in the positive ORP range: $O_2/H_2O$ and $NO_3^-/N_2$ and (ii) anaerobic in the negative ORP range: $SO_4^{2-}/H_2S$ and $CO_2/CH_4$. In the drip environment, since oxygen is readily available, therefore microbes would use $O_2$ as electron acceptor and the dominant redox pair is (1) $O_2/H_2O$ and $NO_3^-/N_2$. However, in the continuous flooded environment, since oxygen is limiting or absent at greater soil depth of at least 15 cm and below where processes such as methanogenesis, methanotrophy, and methylotrophy dominate. There is also an increase in the relative abundance of methanogens and methanotrophs in flooded soil as compared to the drip soil (Fig. 2 to 5). Metagenomic analysis was conducted on duplicate soil samples collected at week 11. All findings and interpretations presented in this study were based on duplicate samples obtained for each rice variety and irrigation treatment group. The experiment was conducted in the greenhouse as it provided precise controls to study the effects of drip irrigation on the rice plants. The soil microbiomes reported in this study also had the same major classes of methanogens and syntrophic bacteria that were present in previous microbiome studies conducted in field trials (48, 49).

The flood environment likely enriches for syntrophic microbial communities that provide more precursors such as acetate, methanol, and formic acid to the methanogens from the breakdown of complex sugars such as cellulose, hemicellulose, and lignin. *Syntrophobacter* and *Syntrophus* were found to be enriched in flooded plots and are not methanogens but rather obligate partners that perform the initial steps of such anaerobic degradation. *Syntrophobacter* species are particularly known for their ability to oxidize propionate into acetate, $CO_2$, and hydrogen or formate. This conversion is only thermodynamically favorable when hydrogen and formate concentrations are kept extremely low, like *Methanobacterium* or *Methanocella* or sulfate-reducing bacteria. Similarly, *Syntrophus* species specialize in the syntrophic degradation of other fatty acids like butyrate and aromatic compounds such as benzoate, also producing acetate, $CO_2$, and hydrogen and formate that are subsequently consumed by their metabolic partners. Both genera are therefore crucial intermediary anaerobes, channeling carbon from more complex organic molecules towards substrates usable by acetoclastic and hydrogenotrophic methanogens, ultimately leading to methane formation in environments like anoxic soils.

## Co-occurrence network reveals functionally distinct hubs of microbes

Molecular ecological network analysis revealed pairwise correlations among methane emissions, ORP measurements, and microbial interactions. Nodes in the top half of the co-occurrence network model in Fig. 8a consist of anaerobes and are aligned with the observed FAPROTAX functional groups with processes such as methanogenesis, methanotrophy, methylotrophy, methanol oxidation, hydrocarbon and aromatic compound degradation enriched in these anaerobic clusters (Fig. S5). Enrichment in methanogenic-related gene functions was also observed higher in flooded soils in the KEGG heatmaps (Fig. S4). Most of these gene functions are related to tetrahydromethanopterin S-methyltransferase and formylmethanofuran dehydrogenase. Both co-occurrence analysis and functional gene annotations are predicted/postulated based on taxonomy and do not necessarily reflect active transcription or metabolic activities. Microbes from these anaerobic clusters in Fig. 8a were highly abundant in continuously flooded environments and are linked to the hydrogen and aromatic compound degradation functions that break down complex substrates such as hemicellulose, cellulose, lignin, and chitin to acetate. Most of these comprise bacteria from the Acetinomycetia class, for example *Actinomadura*, *Dactylosporangium*, *Actinomarinocola*. The methane and methanol cycling cluster showed a highly dense network and was found to correlate closely with methane flux changes. *Methanosarcina* was identified as one of the central nodes of this cluster and likely acts as the central hub in the network. *Methanosarcina* is known to be a generalist and can perform methanogenesis in both acetoclastic and hydrogenotrophic pathways (50). The cellulose decomposition cluster consisted of a complex mix of cellulose degraders like *Cellulomonas* and *Nocardioides*. The syntrophy-driven C-N-S cycling cluster comprises *Syntrophus* and methanogens like *Methanocella*, *Methanoculleus,* and *Methanoregula* alongside sulfur-reducing species like *Desulfobulbus*, *Desulforhabdus,* and sulfur oxidizers like *Thiobacillus* and *Sulfurifustis*.

The cluster associated with methane and methanol cycling cluster consists of taxa such as *Methanosarcina*, *Methylobacterium*, *Micromonospora*, *Phytohabitans*, *Gemmata*, *Hyphomicrobium*, *Rhodococcus*, *Dactylosporangium*, *Nonomuraea,* and *Streptomyces*. This assemblage includes *Methanosarcina*, Actinomycetes like *Dactylosporangium*, *Micromonospora,* and methanotrophs such as *Methylobacterium*. Methanogens actively produce methane under anaerobic conditions, while methanotrophs feed on the methane

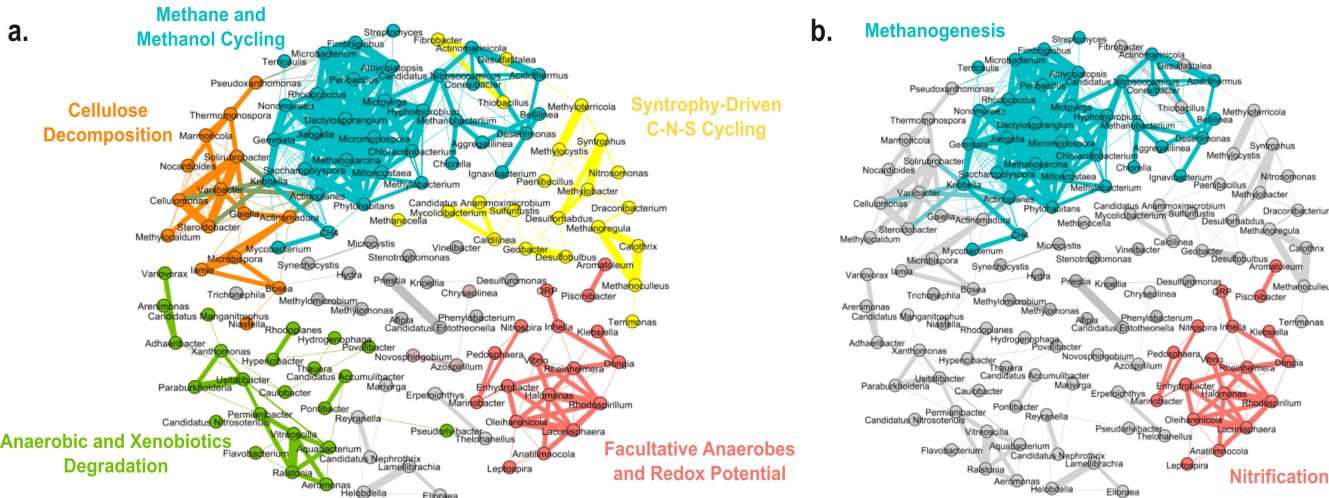

**FIG 8** Co-occurrence network graph of keystone taxa in the top 200 genera. (a) CH$_4$ node is correlated with methane and methanol cycling cluster. ORP node is correlated with facultative anaerobes and redox potential cluster. Five co-occurrence clusters were detected. Each node represents one genus, and an edge is drawn between two nodes if they have a Spearman rank correlation of greater than or equal to 0.6. The nodes are grouped according to clusters based on the modularity algorithm in Gephi and have a modularity index score of 0.671. (b) The network can be simplified into two co-occurrence clusters, where one of the clusters is linked to methanogenesis as reflected by correlation to CH$_4$ fluxes and the other to nitrification as indicated by ORP readings.

produced in the process (51). In addition, *Planctomycete, Gemmata,* and *Actinomycetes* like *Micromonospora*, *Rhodococcus*, *Saccharopolyspora*, *Jiangella,* and *Dactylosporangium* are also members of the cluster. The cellulose decomposition cluster consists of members like *Cellulomonas* and *Nocardioides* alongside other *Actinomycetes,* and they play a critical role as contributors to turnover of complex biopolymers like cellulose, hemicellulose, lignocellulose, pectin, chitin, and keratin (52). *Phytohabitans* synthesize important soil metabolites and have been reported to harbor carbon-acquiring enzyme activities, influencing the soil organic content (53). *Gemmata* strain Wa1-1 was reported to have a functional methylene tetrahydromethanopterin hydrogenase and can influence the rate of methanogenesis (54). *Rhodococcus* is a heterotroph and can compete with methanotrophs to perform methane oxidation in the rice rhizosphere (55). Therefore, microbial members of the methane cluster are found to be higher in relative abundance in the flood-irrigated soil as seen previously in Fig. 2, and they contribute to higher methane emissions.

Nodes in the bottom half of the network consisted of mostly facultative anaerobes and obligate aerobes and seemed to be related to aerobic chemoheterotrophy function. The facultative anaerobe-redox potential cluster was closely associated with higher ORP redox potential, with *Nitrospira* being more abundant in the drip-irrigated soil samples. In addition, the aerobic-xenobiotics degradation cluster likely associates with the aerobic chemoheterotrophy function as the cluster consists of mainly obligate aerobes such as *Hydrogenophaga*, *Arenimonas,* and *Ustatibacter*. The molecular ecological networks showed complex interplay between environmental factors and microbial community dynamics. These functional hubs not only revealed niche-specific microbial interactions but also provided insights into the possible biogeochemical pathways governing methane, nitrogen, and sulfur cycles in the soil ecosystem associated with flood and drip irrigation regimes. The facultative anaerobes - redox potential cluster exhibits a strong positive correlation with higher ORP redox potential readings. The cluster consists of *Nitrospira*, *Aromatoleum*, *Inhella*, *Klebsiella*, *Halomonas*, *Rhodospirillum*, *Lacunisphaera,* and *Marinobacter*. *Nitrospira* plays an important role as an ammonia oxidizer in soil nitrification process (56). *Aromatoleum* utilizes a variety of organic compounds that are linked to denitrification and aerobic respiration (57). *Inhella*, *Halomonas,* and *Klebsiella* have been reported in wastewater treatment studies to be able to perform both simultaneous nitrification and denitrification processes (58–60). Likewise, *Marinobacter,* a marine bacterium, can carry out dissimilatory nitrate reduction, denitrification, and nitrogen metabolism (61). Lastly, *Rhodospirillum* is involved in nitrogen fixation to convert dinitrogen into ammonia (62).

Interestingly, microbes within the syntrophy-driven carbon-nitrogen-sulfur cycling cluster are likely driving the continuous turnover of oxidized and reduced states of carbon, nitrogen, and sulfur in the soil. In the carbon cycle, *Fibrobacter* assists in the digestion of cellulose and is commonly found in the gut of cattle rumen, and *Terrimonas* can secrete extracellular polymeric substance to promote the aggregation of methanogenic communities (63, 64). *Syntrophus* converts organic carbon to acetate and formate, producing hydrogen gas (65). *Methanoculleus*, *Methanocella,* and *Methanoregula* are hydrogenotrophic methanogens that use hydrogen as electron donor and carbon dioxide as a carbon source for the biosynthesis of methane (66). Hydrogen gas produced is then subsequently consumed by methanotrophs like *Methylocystis* and *Methylobacter* (67, 68). *Desulfobulbus*, *Desulfatitalea,* and *Desulforhabdus* are sulfate-reducing bacteria and compete with the hydrogenotrophic methanogens for hydrogen and sugar substrates (69–71). Oxidizers like *Thiobacillus* and *Sulfurifustis* recycle sulfur between its oxidized and reduced states (72, 73). In the nitrogen-related pathways, *Geobacter* is involved in direct interspecies electron transfer, while both *Geobacter* and *Nitrosomonas* oxidize ammonium in soil to help remove nitrogen from soil efficiently (74, 75).

Although the metagenomics data showed the crucial roles of the microorganisms in the five clusters, a part of the microbiome and their functions remains unknown. The co-occurrence analysis could be further simplified into two clusters to identify taxa

that may be involved in either methanogenesis or nitrification activities (Fig. 8b). It was challenging to analyze and separate the biochemical pathways and activities of the different functional guilds in the system as shotgun metagenomics captures a snapshot of the microbiomes and their putative activities at a particular moment in time. Hence, we focused on finding the link between methane emissions and the underlying changes in microbial diversity for the two irrigation systems instead. Our work suggests the need for further study of the longitudinal changes of the microbiome at seedling, maturing, and harvesting stages, as well as the qualitative and quantitative tracking of soil carbon and seasonal variations, if any, with the different irrigation regimes of rice cultivation.

## Conclusions

In summary, there are benefits and trade-offs for continuous flooding and drip irrigation for rice farming. Traditional continuous flooding is effective against weeds. While flooding consumes large volumes of water, it enriches for diverse groups of syntrophic microorganisms that could help promote better plant growth and degrade complex sugars to acetate and methanol precursors favorable for methanogens. The anaerobic environment in floodwater further promotes growth of methanogens that in turn produce more methane. In contrast, drip irrigation is a water-saving technology, and it induces a positive shift in redox potential and enhanced oxygen availability, making it less favorable for methanogens to thrive in. The soil microbiome is an effective way of characterizing the microbial profiles and predicting corresponding microbial activities that may correlate with methane flux changes in the rice field. However, deploying shotgun metagenomics at scale can be slow and expensive. Therefore, it may be more cost-effective to employ direct ORP field measurements and qPCR-based *mcrA* gene assays in the field as a proxy to estimate the level of methanogenic activities of the soil. Our results suggest that drip irrigation can be a potential candidate for future climate-friendly rice farming practices. We showed that the interplay between water management and microbial communities affects methane emissions.

## ACKNOWLEDGMENTS

The authors thank Zhong Chao Yin for sharing the rice varieties and providing helpful suggestions throughout the project. We acknowledge project management support from Phuay Yee Goh and Jeanne Ong. We thank Smitha Chandrasekharan and Gandhi Mathi for help with sample collection for methane analysis. We thank the Axil Scientific (1st BASE) team for metagenomic sequencing and computational analyzes. This research was funded by grants from the Philanthropy Asia Alliance, the Gates Foundation, and the Temasek Life Sciences Laboratory, Singapore.

N.I.N., S.R., and K.J.X.L. designed the experiments and analysis framework. K.J.X.L. performed the soil collection, root imaging, DNA extraction, microbiome data analysis, bioinformatics, and statistical analyses. Methane emissions analysis was performed by A.M. and B.C., A.M. and M.S.T.S. cultivated and maintained the rice plants. N.I.N. supervised K.J.X.L. on both the research work, experimental design, data analyses, and manuscript writing. N.I.N. and K.J.X.L. analyzed the microbiome data and co-wrote the manuscript. N.I.N., S.R., and Zhong Chao Yin were responsible for project funding. All listed authors have read and approved the manuscript.

## AUTHOR AFFILIATIONS

[1]Temasek Life Sciences Laboratory, National University of Singapore, Singapore, Singapore
[2]Department of Biological Sciences, National University of Singapore, Singapore, Singapore

## AUTHOR ORCIDs

Kenny J. X. Lau  http://orcid.org/0000-0002-3666-5253
Naweed I. Naqvi  http://orcid.org/0000-0002-3619-4906

## FUNDING

| Funder | Grant(s) | Author(s) |
|---|---|---|
| Temasek Life Sciences Laboratory | Intramural | Ali Ma |
| | | Bin Chen |
| | | Maria Shibu Thankaraj Salammal |
| | | Srinivasan Ramachandran |
| | | Naweed I. Naqvi |
| Philanthropy Asia Alliance | | Srinivasan Ramachandran |
| | | Naweed I. Naqvi |
| Bill and Melinda Gates Foundation (GF) | | Srinivasan Ramachandran |
| | | Naweed I. Naqvi |

## AUTHOR CONTRIBUTIONS

Kenny J. X. Lau, Data curation, Formal analysis, Investigation, Methodology, Software, Validation, Writing – original draft, Writing – review and editing | Ali Ma, Formal analysis, Investigation, Methodology, Validation | Bin Chen, Formal analysis, Investigation, Methodology, Validation | Maria Shibu Thankaraj Salammal, Formal analysis, Investigation, Methodology, Validation | Srinivasan Ramachandran, Funding acquisition, Project administration, Resources, Writing – review and editing | Naweed I. Naqvi, Conceptualization, Data curation, Funding acquisition, Investigation, Project administration, Resources, Supervision, Validation, Writing – original draft, Writing – review and editing

## DATA AVAILABILITY

Raw metagenomic sequencing data have been submitted to NCBI with the BioProject ID of PRJNA1193822 and accession numbers SRX26959100 to SRX26959107.

The list of differential taxa, metagenome assembled contigs, taxonomic bins, and KEGG annotations in drip and flood-irrigated soil areis available in the supplementary material as separate excel spreadsheets, and the data has been deposited at Dryad: https://datadryad.org/dataset/doi:10.5061/dryad.76hdr7t8s. Bioinformatics analysesanalyzes performed using Kraken2 and Kaiju are outlined in this KBase narrative (https://narrative.kbase.us/narrative/232903).

## ADDITIONAL FILES

The following material is available online.

### Supplemental Material

**Supplemental material (Spectrum02970-25-s0001.docx).** Figures S1 to S8; Tables S1 to S6.

### Open Peer Review

**PEER REVIEW HISTORY (review-history.pdf).** An accounting of the reviewer comments and feedback.

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
