## [Reviewer comments · Microbiology Spectrum]

Microbiology Spectrum

Controlled irrigation suppresses methane emissions by reshaping the rhizosphere microbiomes in rice

Kenny Lau, Ali Ma, Bin Chen, Maria Shibu Thankaraj Salammal, Srinivasan Ramachandran, and Naweed Naqvi

Corresponding Author(s): Naweed Naqvi, Temasek Life Sciences Laboratory Ltd

Review Timeline:

Submission Date:	September 16, 2025
Editorial Decision:	October 21, 2025
Revision Received:	November 17, 2025
Accepted:	November 25, 2025

Editor: Frédérique Reverchon

Reviewer(s): Disclosure of reviewer identity is with reference to reviewer comments included in decision letter(s). The following individuals involved in review of your submission have agreed to reveal their identity: Mirna Vázquez-Rosas-Landa (Reviewer #2)

Transaction Report:

DOI: <https://doi.org/10.1128/spectrum.02970-25>

Re: Spectrum02970-25 (**Controlled irrigation suppresses methane emissions by reshaping the rhizosphere microbiomes in rice**)

Dear Prof. Naweed Isaak Naqvi:

Thank you for the privilege of reviewing your work. Below you will find my comments, instructions from the Spectrum editorial office, and the reviewer comments.

I have now received comments by two independent reviewers, who both highlight the merits of your manuscript but recommend some modifications before it could be accepted for publication in Spectrum. Please find their detailed reviews below.

Revision Guidelines

Sincerely,
Frédérique Reverchon
Editor
Microbiology Spectrum

Reviewer #1 (Comments for the Author):

The manuscript by Lau et al. describes a greenhouse-scale experiment investigating methane emissions and the microbiome supporting two rice varieties cultivated with different irrigation techniques. They observed that rice varieties grown under flooded

irrigation had significantly higher methane emissions than those under drip irrigation. They also used qPCR to track the copy number of the key methanogenesis gene *mcrA* over the same period, with *mcrA* copy number correlating with methane emissions. They employ metagenomic sequencing to identify taxa present at the final timepoint (harvest) of the experiment. They analyze this to determine the taxonomic composition and to hypothesize the functional groups present. This work will be of interest to researchers working in agricultural and wetland systems, and particularly highlights how drip irrigation could lower methane emissions from rice cultivation. Their sequence data is publically available.

Major concerns:

1. The replication of the metagenome sequencing is below microbiome field standard, where triplicate samples is the standard minimum. From what I can gather, there are duplicates of each rice cultivar-irrigation combination. While it is unlikely to be feasible for the authors to repeat the experiment to improve their replication, I would like them to discuss the limitation of their sampling. I would also like them to use more care in presenting the data- for example, in figures 1 and 6 and the supplementary figure of the PCA, please overlay variety. It would also be good to overlay individual data points when using boxplots to visualize the actual data.

2. The manuscript is missing critical supplemental data files to support the evaluation and reproducibility of the analyses. The authors should provide a table detailing the number of reads, the assembly statistics, and the fraction of reads that were assembled. See recent Microbiology Spectrum papers for examples:

a. <https://doi.org/10.1128/spectrum.01291-25>

b. <https://doi.org/10.1128/spectrum.00542-25>

Furthermore, the authors should provide the nucleic acid sequences of their assembled contigs, and all the tables used for microbiome analyses should be provided (for example, "taxonomic tables at genus and class levels", line 194, FAPROTAX output file, line 210-211).

3. The authors should use caution in interpreting their results. qPCR of the *mcrA* gene copy number and metagenome sequencing only report the presence of nucleic acids, not the activity of the microbes or their metabolisms. Furthermore, they may be confounded by relic DNA. The authors should include caution in their discussion about the limitations of their study.

4. The authors should take care in their interpretation of the FAPROTAX data, which is based entirely on taxonomy. I would like to see some discussions of the limitations of this method, and a rewording of lines 446-519 to reflect that these metabolisms are postulated by taxonomy. I would prefer that the authors use their assembled metagenomes to look at their gene annotations. Do you see the predicted functions present in the KEGG annotations of the metagenome? Are there functions associated with drip vs flood irrigation if you analyze the gene data with Maaslin2?

Minor Concerns:

1. In the provided files, Figure S4 is missing.

2. How did the authors ensure they sampled the rhizosphere? Generally, this is accomplished by removing a plant and collecting the soil attached to the roots (example: <https://doi.org/10.1038/s41467-025-57213-x>). I found the description of the constructed soil probes in lines 154-160 to be confusing, and it did not describe sampling in proximity to a root. The authors should clarify their sampling approach, perhaps providing a supplemental image or diagram, and consider whether they really sampled the rhizosphere.

3. Line 153: metagenome samples were taken "during the last week of harvest". Can you edit this language to a specific timepoint that connects to figure 1?

4. The manuscript is missing context for how this data compares to other rice microbiome papers. How does the microbiome here compare to field-scale rice microbiome studies?

5. Line 512- remove the word "has"?

6. Line 512- do you mean methane gas?

7. Lines 335-340: where is this data? Can you add a figure or data reference?

8. Line 473: please remove the word "Active"- you are not measuring activity.

9. Figure S2- please adjust the scale of the y-axes to make it so that all data can be analyzed. Also check the value being plotted- is this read count, not relative abundance? I think this should be RPKM?

10. Lines 319-322: this should be in the methods, and also the KEGG annotations should be a supplementary table.

Reviewer #2 (Comments for the Author):

General Assessment

This is a well-designed study addressing an important question in sustainable rice cultivation: how controlled irrigation influences methane emissions and rhizosphere microbial communities. The authors combine field-based methane flux measurements with metagenomic and qPCR analyses, providing a comprehensive dataset. The work is timely and relevant for both agricultural management and microbial ecology. However, several aspects of the metagenomic analysis and data presentation could be improved to enhance the rigor and clarity of the manuscript.

Major Comments

1. Methane and mcrA measurements

The study reports a strong correlation between methane emissions and mcrA gene copy numbers ($R = 0.77$, $p = 2.3e-10$). It would be useful to explicitly mention in the Results section that mcrA abundance was determined by qPCR, and methane flux was measured via GC-FID (lines 118-179). This connection is a strong point of the work, and the authors might emphasize it further in the Discussion to highlight the value of mcrA as a field-level proxy for methane emissions.

2. Metagenomic sequencing and taxonomic assignment

The manuscript states that an average of 51 million "mapped reads" were obtained (line 264). These reads were mapped to de novo assembled contigs, not reference genomes, this should be clarified.

The metagenomic pipeline relied mainly on Barrnap, RDP classifier, and MEGAN. These tools are standard but relatively limited in scope. I recommend validating taxonomic assignments with alternative methods such as Kraken2, Bracken, or MetaPhlAn, which use broader marker gene sets and can improve taxonomic resolution.

Additionally, since sufficient sequencing depth was achieved, it would be worthwhile to attempt MAG reconstruction (metagenome-assembled genomes). This would allow the authors to identify key methanogenic and nitrifying taxa at genome level and assess their functional potential.

3. Functional and ecological interpretation

The link between community structure and environmental drivers (redox potential, oxygen availability) is well established, but the discussion could be deepened by integrating functional gene data (KEGG/FAPROTAX) with the ecological network results.

Currently, the co-occurrence network (Fig. 8) shows correlations but does not clearly identify functional modules. Consider simplifying the figure to highlight the most relevant clusters (e.g., methanogenesis vs. nitrification).

4. Root morphology measurements

The rationale for including root morphology remains unclear. While the results are interesting, it would help to explain why root traits were expected to influence methane emissions or microbial composition (e.g., via oxygen transport, exudation, or soil structure).

5. Origin of microbial inoculum

The Methods indicate that soil was prepared with topsoil, peat moss, manure, and straw (lines 136-138), but it is not explicitly stated whether microbial communities originated from local soil inoculum. Clarifying this would help interpret how representative the observed communities are of natural paddy soils.

Minor Comments

Figure 4-5 (Krona plots): These figures are visually complex and difficult to interpret. Replacing them with bar plots or phylogenetic trees of relative abundances would improve readability.

Figure 8 (co-occurrence network): The network appears overly dense. Consider focusing on key taxa or subnetworks.

References 7 and 8: Correctly cited as foundational literature on methanogenesis and redox ecology (Conrad, 2020; Zinder, 1993).

Text precision: Clarify that the study used shotgun metagenomics, not a targeted 16S region.

Data richness: The results suggest the data could support additional functional or phylogenetic analyses beyond those currently presented.

Overall Evaluation

This manuscript provides valuable empirical evidence that controlled (drip) irrigation can reduce methane emissions in rice cultivation by altering redox potential and reshaping rhizosphere microbiomes. The coupling of methane flux data with mcrA quantification is particularly strong.

To maximize impact, the metagenomic component should be expanded by improving taxonomic resolution, incorporating genome-resolved analyses, and simplifying figure design.

With these revisions, the study would represent a significant contribution to the fields of microbial ecology, biogeochemistry, and sustainable agriculture.

The manuscript by Lau et al. describes a greenhouse-scale experiment investigating methane emissions and the microbiome supporting two rice varieties cultivated with different irrigation techniques. They observed that rice varieties grown under flooded irrigation had significantly higher methane emissions than those under drip irrigation. They also used qPCR to track the copy number of the key methanogenesis gene *mcrA* over the same period, with *mcrA* copy number correlating with methane emissions. They employ metagenomic sequencing to identify taxa present at the final timepoint (harvest) of the experiment. They analyze this to determine the taxonomic composition and to hypothesize the functional groups present. This work will be of interest to researchers working in agricultural and wetland systems, and particularly highlights how drip irrigation could lower methane emissions from rice cultivation. Their sequence data is publically available.

Major concerns:

1. The replication of the metagenome sequencing is below microbiome field standard, where triplicate samples is the standard minimum. From what I can gather, there are duplicates of each rice cultivar+irrigation combination. While it is unlikely to be feasible for the authors to repeat the experiment to improve their replication, I would like them to discuss the limitation of their sampling. I would also like them to use more care in presenting the data- for example, in figures 1 and 6 and the supplementary figure of the PCA, please overlay variety. It would also be good to overlay individual data points when using boxplots to visualize the actual data.
2. The manuscript is missing critical supplemental data files to support the evaluation and reproducibility of the analyses. The authors should provide a table detailing the number of reads, the assembly statistics, and the fraction of reads that were assembled. See recent *Microbiology Spectrum* papers for examples:
 - a. <https://doi.org/10.1128/spectrum.01291-25>
 - b. <https://doi.org/10.1128/spectrum.00542-25>

Furthermore, the authors should provide the nucleic acid sequences of their assembled contigs, and all the tables used for microbiome analyses should be provided (for example, “taxonomic tables at genus and class levels”, line 194, FAPROTAX output file, line 210-211).

3. The authors should use caution in interpreting their results. qPCR of the *mcrA* gene copy number and metagenome sequencing only report the presence of nucleic acids, not the activity of the microbes or their metabolisms. Furthermore, they may be confounded by relic DNA. The authors should include caution in their discussion about the limitations of their study.

4. The authors should take care in their interpretation of the FAPROTAX data, which is based entirely on taxonomy. I would like to see some discussions of the limitations of this method, and a rewording of lines 446-519 to reflect that these metabolisms are postulated by taxonomy. I would prefer that the authors use their assembled metagenomes to look at their gene annotations. Do you see the predicted functions present in the KEGG annotations of the metagenome? Are there functions associated with drip vs flood irrigation if you analyze the gene data with Maaslin2?

Minor Concerns:

1. In the provided files, Figure S4 is missing.
2. How did the authors ensure they sampled the rhizosphere? Generally, this is accomplished by removing a plant and collecting the soil attached to the roots (example: <https://doi.org/10.1038/s41467-025-57213-x>). I found the description of the constructed soil probes in lines 154-160 to be confusing, and it did not describe sampling in proximity to a root. The authors should clarify their sampling approach, perhaps providing a supplemental image or diagram, and consider whether they really sampled the rhizosphere.
3. Line 153: metagenome samples were taken “during the last week of harvest”. Can you edit this language to a specific timepoint that connects to figure 1?
4. The manuscript is missing context for how this data compares to other rice microbiome papers. How does the microbiome here compare to field-scale rice microbiome studies?
5. Line 512- remove the word “has”?
6. Line 512- do you mean methane gas?
7. Lines 335-340: where is this data? Can you add a figure or data reference?
8. Line 473: please remove the word “Active”- you are not measuring activity.
9. Figure S2- please adjust the scale of the y-axes to make it so that all data can be analyzed. Also check the value being plotted- is this read count, not relative abundance? I think this should be RPKM?
10. Lines 319-322: this should be in the methods, and also the KEGG annotations should be a supplementary table.

Point-by-Point Response to Reviewers' comments

Reviewer #1 (Comments for the Author):

The manuscript by Lau et al. describes a greenhouse-scale experiment investigating methane emissions and the microbiome supporting two rice varieties cultivated with different irrigation techniques. They observed that rice varieties grown under flooded irrigation had significantly higher methane emissions than those under drip irrigation. They also used qPCR to track the copy number of the key methanogenesis gene *mcrA* over the same period, with *mcrA* copy number correlating with methane emissions. They employ metagenomic sequencing to identify taxa present at the final timepoint (harvest) of the experiment. They analyze this to determine the taxonomic composition and to hypothesize the functional groups present. This work will be of interest to researchers working in agricultural and wetland systems, and particularly highlights how drip irrigation could lower methane emissions from rice cultivation. Their sequence data is publically available.

Major concerns:

1. The replication of the metagenome sequencing is below microbiome field standard, where triplicate samples is the standard minimum. From what I can gather, there are duplicates of each rice cultivar+irrigation combination. While it is unlikely to be feasible for the authors to repeat the experiment to improve their replication, I would like them to discuss the limitation of their sampling.

Many thanks for the positive response to our manuscript and for highlighting the importance, broader interests, and the open access availability of the sequence data. We agree that triplicate sampling is the required norm, and have added a couple sentences (Lines 475-477) highlighting the limitations of our sampling strategy, which only used duplicates of each rice cultivar+irrigation combination.

I would also like them to use more care in presenting the data- for example, in figures 1 and 6 and the supplementary figure of the PCA, please overlay variety. It would also be good to overlay individual data points when using boxplots to visualize the actual data.

Many thanks for the suggestion, we have now overlaid the individual data points as dots in the boxplot for **Figure 1a** (methane emissions for 2 varieties). Each timepoint consists of 3 replicates.

We have overlaid the variety information within the CCA plot in **Figure 6**. Likewise, the varietal information has been overlaid in Supplementary **Figure S1**. The requisite details have been added in the text in lines 306-309 of the revised manuscript.

The revised figures have been appended below for easy referral and perusal.

2. The manuscript is missing critical supplemental data files to support the evaluation and reproducibility of the analyses. The authors should provide a table detailing the number of reads, the assembly statistics, and the fraction of reads that were assembled. See recent Microbiology Spectrum papers for examples: a. <https://doi.org/10.1128/spectrum.01291-25> b. <https://doi.org/10.1128/spectrum.00542-25>.

Following was the Supplementary Fig S4 that was missing in the combined PDF converted file. This has now been reinstated as Figure S7 in the revised manuscript.

Many thanks for this suggestion. The binning data and contigs information have now been provided in Supplementary data as Excel Files that include the number of reads, the assembly statistics, and the fraction of reads that were assembled. Table S3. Bin_Table.xlsx and Table S4. Contig_Table.xlsx.

Sample	Total Reads	Mapped Reads
HHZ-Drip1	83855896	51915900
HHZ-Drip2	89557428	56319641
HHZ-Flood1	102142512	57194581
HHZ-Flood2	87735596	47905649
TR-Drip1	87984006	56347321
TR-Drip2	80312940	53618116
TR-Flood1	83163652	45950964
TR-Flood2	73536106	40161042

Furthermore, the authors should provide the nucleic acid sequences of their assembled contigs, and all the tables used for microbiome analyses should be provided (for example, "taxonomic tables at genus and class levels", line 194, FAPROTAX output file, line 210-211).

We have provided both the genera and class level taxonomic tables as FASTA and Excel files in the revised manuscript.

Table S5. Genus_Abundance.xlsx

Table S6. Class_Abundance.xlsx

Faprotax file is also available.

Table S7. Faprotax_function.xlsx

Link to Supplementary material at Dryad repository:

http://datadryad.org/share/LINK_NOT_FOR_PUBLICATION/RfqHNlnYecvI2YBLWqYyxMNVT7hUR314X310PBDnW54

Or

<https://datadryad.org/dataset/doi:10.5061/dryad.76hdr7t8s>

3. The authors should use caution in interpreting their results. qPCR of the *mcrA* gene copy number and metagenome sequencing only report the presence of nucleic acids, not the activity of the microbes or their metabolisms. Furthermore, they may be confounded by relic DNA. The authors should include caution in their discussion about the limitations of their study.

We agree with Reviewer 1 that qPCR of DNA extracted from soil only reveals the gene copy number and is not a proxy for measuring metabolic activities of the microbes *per se*. We have included the term gene copy number and this cautionary note in the discussion section (Lines 427-432) as follows: *Our results showed strong correlation between mcrA gene copy number abundance and methane emissions (R = 0.77), the dispersion of data*

points around the regression line suggested substantial variability across individual samples and data points. While *mcrA* gene abundance (qPCR) and metagenomic sequencing served as a useful proxy to detect methanogens, such DNA-based methods do not report the metabolic activity of the microbes per se.

4. The authors should take care in their interpretation of the FAPROTAX data, which is based entirely on taxonomy. I would like to see some discussions of the limitations of this method, and a rewording of lines 446-519 to reflect that these metabolisms are postulated by taxonomy. I would prefer that the authors use their assembled metagenomes to look at their gene annotations. Do you see the predicted functions present in the KEGG annotations of the metagenome? Are there functions associated with drip vs flood irrigation if you analyze the gene data with Maaslin2?

Please note that the heatmap results in revised Fig S5 (previously S3) were plotted based on predicted gene functions in the KEGG annotation analyses. Additionally, we had referred to both FAPROTAX and KEGG pathways in our analyses. Unfortunately, due to the missing Figure S4 in our submission, the Reviewer was unable to see the FAPROTAX results. Fig S4, now presented as Fig S7 in the revised manuscript, has now been provided to address this concern. We have also included the limitation (504-506) that the metabolic functions are predicted/postulated based on taxonomy and do not necessarily reflect active transcription or metabolic activities. Methanogenic-function related genes are higher in Flood as compared to Drip samples using the KEGG analysis. Most of these

gene functions are related to tetrahydromethanopterin S-methyltransferase and formylmethanofuran dehydrogenase.

KEGG annotation file: KEGG_RPKM.csv is provided as supplementary data for reference.

Maaslin2 combined with KEGG provides broad predictions of gene functions, though our focus is limited to archaea, where methanogens constitute less than 1% of the total microbiome. Many of the inferred functions may pertain to soil rhizosphere bacteria. From the KEGG and Maaslin2 analyses, we observed greater upregulation of nutrient transport pathways under flood conditions, whereas drip irrigation was associated with increased expression of genes involved in cellular repair and stress response(s). No significant functional differences were detected between the Huanghuazhan and Temasek rice varieties.

Minor Concerns:

1. In the provided files, Figure S4 is missing.

Following is the Fig S4 that was inadvertently lost during the PDF conversion stage of the submission process. This Figure S4 is now included as S7 in the revised manuscript.

2. How did the authors ensure they sampled the rhizosphere? Generally, this is accomplished by removing a plant and collecting the soil attached to the roots (example: <https://doi.org/10.1038/s41467-025-57213-x>). I found the description of the constructed soil probes in lines 154-160 to be confusing, and it did not describe sampling in proximity to a root. The authors should clarify their sampling approach, perhaps providing a supplemental image or diagram, and consider whether they really sampled the rhizosphere.

We have added the requisite information in the Materials and Methods of the revised version (Lines 157 – 163). Soil samples were collected using av25 mL serological pipette strips that were modified as soil probes (Fig S1a). The serological pipettes were cut using a heat-sterilized knife at the top and between the 16-and 20-mL mark to create an opening at the top and the side (Fig S1b and c). A new 10 mL serological pipette strip was then inserted from the top of the 25 mL pipette strips to enclose the 25 mL strip (Fig S1d and e).

The soil probe was then plunged into the soil adjacent to the stem of the rice plant at a depth of 15 to 20 cm (Fig S1f). The 10 mL strip was then released so that soil can enter the 25 mL pipette strip from the side opening. Soil samples were then collected in a 50 mL tube, transported to the laboratory, and kept frozen at -20°C freezer until DNA extraction.

3. Line 153: metagenome samples were taken "during the last week of harvest". Can you edit this language to a specific timepoint that connects to figure 1?

Samples for metagenomics analysis were collected at Week 11 post-transplantation of rice seedlings. Line 153 has been changed accordingly.

4. The manuscript is missing context for how this data compares to other rice microbiome papers. How does the microbiome here compare to field-scale rice microbiome studies?

Taxa detected in our results show similarity to other studies where soil microbiome analyses were performed and sampled from rice fields. (Line 479) For example, we found *Methanocella*, *Methanosarcina*, *Methanobacterium* and *Methanothrix* as major methanogens as well as *Syntrophus* and *Syntrophobacter* species in the greenhouse study similar to their abundance/presence in rice fields.

Liechty, Z., Santos-Medellín, C., Edwards, J., Nguyen, B., Mikhail, D., Eason, S., Phillips, G. and Sundaresan, V., 2020. Comparative analysis of root microbiomes of rice cultivars with high and low methane emissions reveals differences in abundance of methanogenic archaea and putative upstream fermenters. *mSystems*, 5(1): 101-128.

Li, D., Ni, H., Jiao, S., Lu, Y., Zhou, J., Sun, B. and Liang, Y., 2021. Coexistence patterns of soil methanogens are closely tied to methane generation and community assembly in rice paddies. *Microbiome*, 9(1): 20.

5. Line 512- remove the word "has"?

Thanks, we have removed “has” from Line 512.

6. Line 512- do you mean methane gas?

Line 512 refers to the hydrogenotrophic methanogens that utilize hydrogen as an electron donor and carbon dioxide as a carbon source to carry out chemosynthetic processes.

7. Lines 335-340: where is this data? Can you add a figure or data reference?

Fig S4 was likely lost during PDF conversion in the submission process. We have added Fig S4 as revised S7 now.

8. Line 473: please remove the word "Active"- you are not measuring activity.

We removed “active” from Line 473.

9. Figure S2- please adjust the scale of the y-axes to make it so that all data can be analyzed. Also check the value being plotted- is this read count, not relative abundance? I think this should be RPKM?

We plotted Figure S2 as normalized read count in the Y-axis and rescaled Y-axis to log₁₀ scale. This is useful in rendering the less abundant reads to be more visible.

10. Lines 319-322: this should be in the methods, and also the KEGG annotations should

be a supplementary table.

Many thanks. We moved Lines 319-322 to the Methods section. We have also provided the KEGG annotations as a Supplementary Table: KEGG_RPKM.csv.

Reviewer #2 (Comments for the Author):

General Assessment

This is a well-designed study addressing an important question in sustainable rice cultivation: how controlled irrigation influences methane emissions and rhizosphere microbial communities. The authors combine field-based methane flux measurements with metagenomic and qPCR analyses, providing a comprehensive dataset. The work is timely and relevant for both agricultural management and microbial ecology. However, several aspects of the metagenomic analysis and data presentation could be improved to enhance the rigor and clarity of the manuscript.

Major Comments

1. Methane and *mcrA* measurements

The study reports a strong correlation between methane emissions and *mcrA* gene copy numbers ($R = 0.77$, $p = 2.3e-10$). It would be useful to explicitly mention in the Results section that *mcrA* abundance was determined by qPCR, and methane flux was measured via GC-FID (lines 118-179). This connection is a strong point of the work, and the authors might emphasize it further in the Discussion to highlight the value of *mcrA* as a field-level proxy for methane emissions.

Many thanks for the encouraging remarks. We agree with the Reviewers (1 and 2) and have now clearly stated the limitations and caution about *mcrA* abundance being based on qPCR analysis only. While the correlation is at 0.77, it is still difficult to correlate a strong linear relationship between *mcrA* abundance and methane emissions. Nonetheless, *mcrA* gene abundance is still useful as a proxy to detect methanogens using qPCR-based methods. For instance, Qi et al. (2021) found and showed that *mcrA* gene abundance correlates strongly with methane emissions and biochar also suppresses *mcrA* levels and is shown to reduce methane. [Qi, L., Ma, Z., Chang, S.X., Zhou, P., Huang, R., Wang, Y., Wang, Z. and Gao, M., 2021. Biochar decreases methanogenic archaea abundance and methane emissions in a flooded paddy soil. *Science of the Total Environment*, 752: 141958].

2. Metagenomic sequencing and taxonomic assignment

The manuscript states that an average of 51 million "mapped reads" were obtained (line 264). These reads were mapped to de novo assembled contigs, not reference genomes, this should be clarified.

The metagenomic pipeline relied mainly on Barrnap, RDP classifier, and MEGAN. These

tools are standard but relatively limited in scope. I recommend validating taxonomic assignments with alternative methods such as Kraken2, Bracken, or MetaPhlAn, which use broader marker gene sets and can improve taxonomic resolution.

Additionally, since sufficient sequencing depth was achieved, it would be worthwhile to attempt MAG reconstruction (metagenome-assembled genomes). This would allow the authors to identify key methanogenic and nitrifying taxa at genome level and assess their functional potential.

Many thanks. We have now provided the Contigs Table and Bin Table as supplementary material. We have assembled/reconstructed the contigs into MAGs using Megahit and binned the MAGs using Concoct and optimised using DAS tool. We have 271 binned contigs in total. Among the 271 binned contigs, concoct.119 and 137 showed close alignment with methanogenic archaea. Please refer to Supplementary data, Bin_Table.xls The longest contig in concoct.119 bin is megahit_6813907 with a size of 3737bp. The longest contig in concoct.137 bin is megahit_7806624 with a fragment length of 6180 bp. It is difficult to say that they are mapped to specific functions in the genome.

concoct.119	DAS	k_Archaea;p_Euryarchaeota;c_Methanomicrobia;o_Methanotrichales;f_Methanotrich
concoct.137	DAS	k_Archaea;p_Euryarchaeota;c_Methanomicrobia;

We have provided the Fasta sequence files (Concoct_Fasta_Files.zip) for these 2 bins in the revised manuscript.

As per Reviewer's recommendation, we reanalysed the datasets using Kraken2 and Kaiju. Reads associated with archaeal taxa accounted for less than 1% of the total microbiome. The majority of metagenome-assembled genomes (MAGs) were assigned to bacterial phyla, including Myxococcota, Planctomycetota, Patescibacteria, Actinomycetota, Cyanobacteriota and Pseudomonadota.

As suggested by reviewer 2, we also repeated taxonomic classification with other tools like Kaiju and Kraken2. We get similar percentages of classified reads as output and taxonomic assignments with what was presented with Diamond and the NCBI NR database.

In addition, a narrative on the bioinformatics analyses for both Kaiju and Kraken2 and also the pipeline to create metagenome assembled genomes (MAGs) has been included. We also reperformed analyses from samples to MAGs for these 8 samples on KBase for clarity. Details can be found in the following link at <https://narrative.kbase.us/narrative/232903>.

3. Functional and ecological interpretation

The link between community structure and environmental drivers (redox potential, oxygen availability) is well established, but the discussion could be deepened by integrating functional gene data (KEGG/FAPROTAX) with the ecological network results. Currently, the co-occurrence network (Fig. 8) shows correlations but does not clearly identify functional modules. Consider simplifying the figure to highlight the most relevant clusters (e.g., methanogenesis vs. nitrification).

Many thanks. We have added Fig 8b to simplify the clusters further into meaningful associations that we find with CH₄ and nitrification.

4. Root morphology measurements

The rationale for including root morphology remains unclear. While the results are interesting, it would help to explain why root traits were expected to influence methane emissions or microbial composition (e.g., via oxygen transport, exudation, or soil structure).

Yes, root structure and morphology are important to compare oxygen transport and how the rice plant responds to water under drip irrigation as compared to continuous flooding.

This part has been moved to the Discussion section: **Methane emission profiles in rice soils are associated with the soil microbiome and water management practices**

Changes in root morphology in drip-irrigated rice plants in Fig S7 could be attributed to the plant's response to water availability. In drip irrigation, water is supplied in a controlled manner directly to the soil near the plant roots, reducing the need for extensive root exploration to access water. Our results suggest that the drip-irrigated plant roots tend to grow closer to the soil surface as water becomes scarce with increasing depth. The roots also tend to be thicker near the soil surface, which is likely an adaptation to absorb more moisture under such controlled irrigation regimes. On the other hand, the flood-irrigated rice plants seemed to develop longer and deeper roots with fine lateral growth. It is likely that the plant invests in a deeper root system to anchor effectively and to absorb more nutrients via the active microbial communities under excess water.

5. Origin of microbial inoculum

The Methods indicate that soil was prepared with topsoil, peat moss, manure, and straw (lines 136-138), but it is not explicitly stated whether microbial communities originated from local soil inoculum. Clarifying this would help interpret how representative the observed communities are of natural paddy soils.

Organic matter such as straw and manure are likely potential sources for methanogens and methanotrophs in our greenhouse study. However, the taxa detected in our results show significant similarity to other studies where soil microbiome analyses were performed and sampled from rice fields. For example, we found *Methanocella*, *Methanosarcina*,

*Methanobacterium* and *Methanothrix* as major methanogens as well as *Syntrophus* and *Syntrophobacter* species in the greenhouse study similar to the ones found in rice fields.

Liechty, Z., Santos-Medellín, C., Edwards, J., Nguyen, B., Mikhail, D., Eason, S., Phillips, G. and Sundaresan, V., 2020. Comparative analysis of root microbiomes of rice cultivars with high and low methane emissions reveals differences in abundance of methanogenic archaea and putative upstream fermenters. *Msystems*, *5*(1), pp.10-1128.

Li, D., Ni, H., Jiao, S., Lu, Y., Zhou, J., Sun, B. and Liang, Y., 2021. Coexistence patterns of soil methanogens are closely tied to methane generation and community assembly in rice paddies. *Microbiome*, *9*(1), p.20.

Minor Comments

Figure 4-5 (Krona plots): These figures are visually complex and difficult to interpret. Replacing them with bar plots or phylogenetic trees of relative abundances would improve readability.

Figure 2 and 3 present the taxa as stacked bar charts. However, we felt that Krona plots provide a better representation for Archaea as percentage change in a pie chart where the proportional change in methanogens vs Nitrososphaera is easy to depict and visualize based simply on colour (light brown vs pinkish red). Bar charts can be used (please refer to the figure below), however, the relative abundance is in the range of 0.01 - 1% and makes it difficult to visualize or interpret the data.

Figure 8 (co-occurrence network): The network appears overly dense. Consider focusing on key taxa or subnetworks.

This has been addressed before for Reviewer's Major Points #3 above. In addition, we provided Panel 8b to highlight only the methanogenesis and nitrification networks to make it simple, clear and less cluttered.

References 7 and 8: Correctly cited as foundational literature on methanogenesis and redox ecology (Conrad, 2020; Zinder, 1993).

Yes, we will edit References 7 and 8 to this format.

Text precision: Clarify that the study used shotgun metagenomics, not a targeted 16S region.

We have now clarified (Line 189) that the study used shotgun metagenomics, and did not target the 16S region. Sequence adapters were trimmed from all paired end reads using bbdduk of the BBTools packages [17]. MultiQC [18] was performed and reads with at least a quality Phred score of 20 were filtered. Shotgun metagenomics analysis was performed for the samples. The filtered reads were then assembled *de novo* using Megahit [19] into contigs of at least 500 bp.

Data richness: The results suggest the data could support additional functional or phylogenetic analyses beyond those currently presented.

We have now provided the KEGG annotation file in the revised manuscript. Experiments in this study were done using DNA-based metagenomics analyses. We have included the limitations about related to postulated functions based on KEGG rpkms and Faprotax predictions. We observed an increase in abundance in methanogenic, sulfur-reducing gene functions in flood irrigated soils. Drip irrigation has some enrichment in periplasmic nitrate reductase genes.

Overall Evaluation

This manuscript provides valuable empirical evidence that controlled (drip) irrigation can reduce methane emissions in rice cultivation by altering redox potential and reshaping rhizosphere microbiomes. The coupling of methane flux data with *mcrA* quantification is particularly strong.

To maximize impact, the metagenomic component should be expanded by improving taxonomic resolution, incorporating genome-resolved analyses, and simplifying figure design.

With these revisions, the study would represent a significant contribution to the fields of microbial ecology, biogeochemistry, and sustainable agriculture.

Many thanks for the overall positive response to our submission, and we would like to thank both the reviewers and the Editor for their constructive critique and help in significantly improving the analyses and the manuscript.

Re: Spectrum02970-25R1 (**Controlled irrigation suppresses methane emissions by reshaping the rhizosphere microbiomes in rice**)

Dear Prof. Naweed Isaak Naqvi:

The comments made by both reviewers were addressed and the manuscript can now be accepted for publication in Microbiology Spectrum.

Your manuscript has been accepted, and I am forwarding it to the ASM production staff for publication. Your paper will first be checked to make sure all elements meet the technical requirements. ASM staff will contact you if anything needs to be revised before copyediting and production can begin. Otherwise, you will be notified when your proofs are ready to be viewed.

Sincerely,
Frédérique Reverchon
Editor
Microbiology Spectrum